# Restoration of visual function by transplantation of optogenetically engineered photoreceptors

Marcela Garita-Hernandez[1,6], Maruša Lampič[1,6], Antoine Chaffiol[1], Laure Guibbal[1], Fiona Routet[1], Tiago Santos-Ferreira[2], Sylvia Gasparini[2], Oliver Borsch[2], Giuliana Gagliardi[1], Sacha Reichman [1], Serge Picaud [1], José-Alain Sahel[1,3,4], Olivier Goureau[1], Marius Ader[2], Deniz Dalkara[1,7] & Jens Duebel[1,5,7]

A major challenge in the treatment of retinal degenerative diseases, with the transplantation of replacement photoreceptors, is the difficulty in inducing the grafted cells to grow and maintain light sensitive outer segments in the host retina, which depends on proper interaction with the underlying retinal pigment epithelium (RPE). Here, for an RPE-independent treatment approach, we introduce a hyperpolarizing microbial opsin into photoreceptor precursors from newborn mice, and transplant them into blind mice lacking the photoreceptor layer. These optogenetically-transformed photoreceptors are light responsive and their transplantation leads to the recovery of visual function, as shown by ganglion cell recordings and behavioral tests. Subsequently, we generate cone photoreceptors from human induced pluripotent stem cells, expressing the chloride pump Jaws. After transplantation into blind mice, we observe light-driven responses at the photoreceptor and ganglion cell levels. These results demonstrate that structural and functional retinal repair is possible by combining stem cell therapy and optogenetics.

[1] Sorbonne Université, Institut de la Vision, INSERM, CNRS, 75012 Paris, France. [2] CRTD/Center for Regenerative Therapies Dresden, CMCB, TU Dresden, Dresden, Germany. [3] CHNO des Quinze—Vingts, DHU Sight Restore, INSERM-DGOS CIC 1423, Paris, France. [4] Department of Ophthalmology, The University of Pittsburgh School of Medicine, Pittsburgh, USA. [5] Department of Ophthalmology, University Medical Center Göttingen, Göttingen, Germany. [6] These authors contributed equally: Marcela Garita-Hernandez, Maruša Lampič. [7] These authors jointly supervised this work: Deniz Dalkara, Jens Duebel. Correspondence and requests for materials should be addressed to D.D. (email: deniz.dalkara@gmail.com) or to J.D. (email: jens.duebel@gmail.com)

Cell replacement therapy offers hope for the treatment of late-stage retinal degeneration, when the outer retinal photoreceptor layer is lost[1–3]. However, a remaining obstacle of photoreceptor replacement is that transplanted cells have to develop into functional photoreceptors with light sensitive outer segments (OS). Indeed, in mouse models of severe degeneration, the formation of light-sensitive OS by transplanted photoreceptors has been difficult to achieve[4–6]. Recent studies, using retinal sheet transplantation lead to major improvements in terms of OS formation and light sensitivity[7,8]. Despite these promising results, a major problem has not yet been solved: photoreceptors need tight interaction with the retinal pigment epithelium (RPE) in order to maintain their structure and function via continuous disc shedding and renewal[9]. Since in retinal degenerative diseases the RPE is often also compromised[9,10], the probability that transplanted photoreceptors stay sensitive to light is very low[11,12]. To tackle this problem, we introduce optogenetic light sensors into photoreceptors, derived from the developing mouse retina as well as from human induced pluripotent stem cells (hiPSCs), and transplant them into mouse models of severe retinal degeneration. The key point of our approach is that these optogenetically-transformed photoreceptors stay functional based on the activity of the microbial opsin, even in the absence of properly formed OS and without the support from the RPE.

## Results

**Neonatal mouse-derived NpHR photoreceptor precursors.** For optogenetic transformation of mouse photoreceptors, eyes of newborn wild-type mice at post-natal day (P) 2 were injected with an adeno-associated viral (AAV) vector encoding enhanced *Natronomonas pharaonis* halorhodopsin eNpHR2.0 (NpHR)[13] under the control of the rhodopsin promoter (AAV-Rho-NpHR-YFP) (Fig. 1a and Supplementary Fig. 1). At P4, photoreceptor precursors were sorted by magnetic activated cell sorting (MACS) using the photoreceptor specific cell surface marker CD73[14,15]. The harvested cells were transplanted via sub-retinal injections into two blind mouse models of late-stage retinal degeneration (*Cpfl1/Rho*$^{-/-}$ mice[16] aged 9 to 18 weeks and *C3H rd/rd* (rd1) mice[17] aged 4 to 11 weeks; see Supplementary Table 1 for a complete overview of mouse ages). At these ages, the vast majority of outer nuclear layer (ONL) cells were lost in host mice (Fig. 1b, e). *Cpfl1/Rho*$^{-/-}$ mice are left with 2–3 rows of photoreceptors at the age of 9 weeks, and a single row of photoreceptors by 10–12 weeks of age. These mice are born with non-functional rods and cones[16]. Rd1 mice loose photoreceptor OS and only a single row of cone cell bodies in the ONL remains by 3 weeks after birth[18]. Four weeks after transplantation, we investigated the morphology of the transplanted donor cells and their ability to integrate into the host retina. In both mouse models, we found NpHR-positive donor cells in close contact to cell bodies of rod bipolar cells, but none of the transplanted cells displayed correctly formed OS (Fig. 1c, d, f, g). Transplanted cells expressed the synaptic marker Synaptophysin (Supplementary Fig. 2) suggesting synapse formation between donor photoreceptors and the downstream neurons. We quantified the number of YFP$^+$ cells in the subretinal space transplanted with donor-derived NpHR-expressing rod precursors and found substantial numbers of cells to survive at four weeks post transplantation (Supplementary Fig. 3). Next, we assessed potential material transfer between transplanted cells and remaining photoreceptors by fluorescence in situ hybridization with Y chromosome-specific probe (Y chromosome FISH). NpHR-expressing rod precursors derived from male P4 mice were injected into female *Cpfl1/Rho*$^{-/-}$ mice at 9 weeks of age, and imaged after 4 weeks using structured illumination microscopy (Fig. 1h). Y chromosome$^+$/YPF$^+$ cells

(transplanted donor cells) and Y chromosome$^-$/YFP$^+$ cells (endogenous photoreceptor that underwent material transfer) were quantified. 90% of YFP$^+$ cells co-stained with Y chromosome probe, leaving only very few cells exclusively YFP$^+$. This could either be due to an artifact or very rare events of cytoplasmic exchange among donor and host photoreceptors (Fig. 1h, i). We then tested if we can elicit light responses from these NpHR-positive donor cells in the absence of functional OS. Two-photon targeted patch-clamp recordings revealed robust responses to orange light pulses (580 nm, $10^{16}$ photons cm$^{-2}$s$^{-1}$) (Fig. 2b and Supplementary Fig. 4). There were no measurable light-evoked currents in transplanted photoreceptors expressing GFP only, which is consistent with the finding that the transplanted cells lacked their light sensitive OS. Stimulation at different wavelengths showed a spectral sensitivity matching the action spectrum of NpHR (Fig. 2b). To measure the temporal properties of NpHR-positive photoreceptors, we recorded photocurrents using light pulses at increasing frequencies, and we observed that they could follow up to 25 Hz (Fig. 2c and Supplementary Fig. 4). Although, frequencies above 10 Hz are filtered out by the bipolar cells, the ability of optogenetically engineered photoreceptors to respond to light in a faster than natural pace implies that retinal ganglion cells (RGCs) receiving signal from these cells should follow high-frequency stimulation in a similar manner to normal retina[19]. The rise constants were significantly faster compared to photocurrents of wild-type mice (Fig. 2d). Both, from the spectral (peak current at 580 nm) and the temporal (Tau$_{ON}$ < 10 ms) response properties we concluded that the photocurrents were driven by the introduced NpHR (Fig. 2a–d and Supplementary Fig. 4).

**Connectivity and signal transmission to host neurons.** Next, we investigated if the signals from transplanted photoreceptors are transmitted to RGCs, the output neurons of the retina. By using extracellular spike recordings, we measured ON- and OFF-light responses in RGCs. These results demonstrate that NpHR-induced signals are transmitted to the retinal output neurons via ON- and OFF-pathways suggesting that the transplanted photoreceptors can form functional synaptic connections with the inner retinal neurons (Fig. 3a and Supplementary Fig. 5), which was supported by histological analysis (Supplementary Fig. 2). Recordings performed under pharmacological block of photoreceptor input to ON-bipolar cells (50 μM L-AP4) showed complete abolition of ON light responses, which recovered after 20 min of L-AP4-washout. These control experiments confirmed that light-induced signals were indeed transmitted via photoreceptor-to-bipolar cell synapses (Fig. 3b, c). By stimulating treated retinas at different wavelengths we determined the spectral sensitivity of the light responses, which peaked at 580–600 nm, reflecting the action spectrum of NpHR (Fig. 3d, e). To assess the light intensities required to trigger spike responses, we used light pulses (580 nm) at different intensities. Importantly, the intensities required to evoke light responses were well below the safety limit for optical radiation in the human eye[20,21] (Fig. 3f, g and Supplementary Fig. 5). We did not observe measurable light responses in retinae from age-matched control-mice, where photoreceptor precursors expressing only GFP were transplanted (Fig. 3h, i and Supplementary Fig. 5). Lastly, to test whether the behaviour of treated mice could be modulated by light, we used the light/dark box test[22] employing high intensity orange light (Fig. 3j). Treated *Cpfl1/Rho*$^{-/-}$ mice displayed robust light avoidance behaviour (40.7 ± 3.5% of time in the illuminated compartment), compared to non-injected (59.8 ± 2.2%) mice and mice transplanted with photoreceptor precursors expressing GFP (56.6 ± 4.5%) (Fig. 3k).

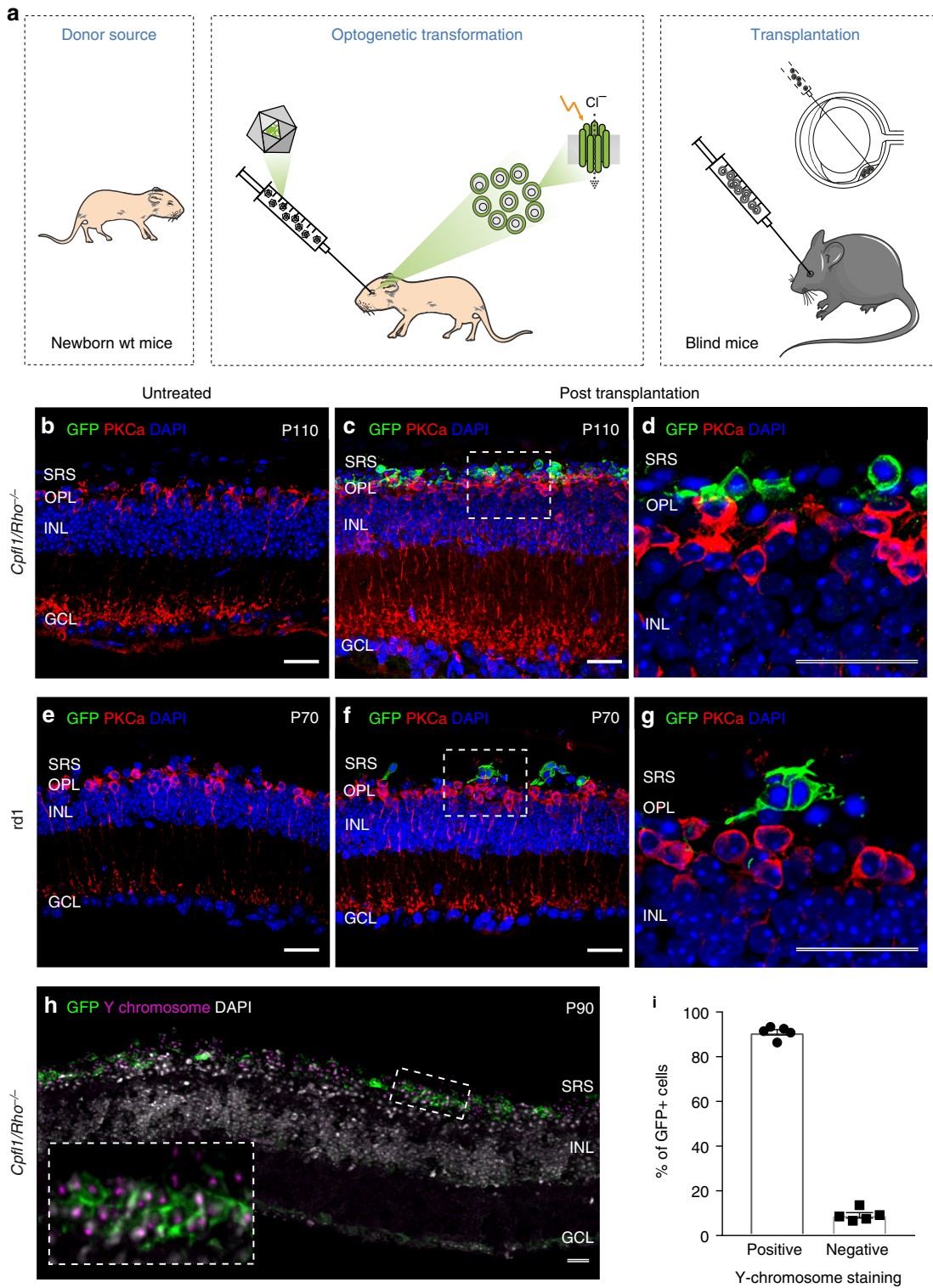

**Generation of hiPSC-derived Jaws-expressing photoreceptors**. To evaluate the translatability of our approach to human subjects, we asked if it is possible to replace the mouse donor cells with optogenetically-transformed hiPSCs (Fig. 4a). To do so, we first optimized a previous protocol of differentiation based on the self-generation of 3D neural-retina-like structures[23]. Using this system, we generated cone-enriched retinal organoids, expressing the pan-photoreceptor markers Cone Rod Homeobox (CRX) and recoverin (RCVRN) alongside the cone-specific marker cone arrestin (CAR) (Fig. 4b–f and Supplementary Fig. 6). Contrary to nocturnal rodents, cone photoreceptors are responsible for high acuity daylight vision in humans, and are therefore the preferred choice for transplantation. To render these immature cones light sensitive, we used the hyperpolarizing chloride pump Jaws, a red-shifted cruxhalorhodopsin, Jaws, derived from *Haloarcula (Halobacterium) salinarum* and engineered to result in red light–induced photocurrents three times those of earlier silencers[24]. Jaws was chosen for iPSC experiments based on its enhanced expression level and improved membrane trafficking in human tissue, compared to NpHR[24–26]. By using an AAV vector,

**Fig. 1** Transplanted photoreceptor precursors, expressing NpHR, integrate into the retina of blind mice. **a** Eyes of wild-type mice at P2 were injected with AAV-Rho-NpHR-YFP. Two days later, retinas were dissected and photoreceptor precursors sorted out. These cells were transplanted via sub-retinal injections into blind mice. **b–g** Immunofluorescence analysis on vertical sections of $Cpfl1/Rho^{-/-}$ (**b–d**) and rd1 (**e–g**) retinas. **b** Age-matched non-transplanted $Cpfl1/Rho^{-/-}$ retina. **c, d** $Cpfl1/Rho^{-/-}$ retina transplanted with NpHR-photoreceptors showing NpHR-YFP$^+$ cells (stained with anti-GFP antibody, green) located on top of host PKCα bipolar cells (red). **e** Age-matched non-transplanted rd1 retina. **f, g** Rd1 retina transplanted with NpHR-photoreceptors. **h, i** Y chromosome FISH. **h** A retinal section showing Y chromosome labelling (magenta) and immunohistochemistry staining for GFP (green) with DAPI counterstaining (white) 4 weeks after transplantation of NpHR-expressing rods from male donors into a female $Cpfl1/Rho^{-/-}$ mouse (P60 at the time of transplantation). **i** Quantification of NpHR-expressing cells containing Y chromosome from five individual experimental retinas ($N = 5$). The vast majority of NpHR-YFP$^+$ cells (stained with anti-GFP antibody) also contained a Y chromosome (90.9 ± 1.2%), proving that they originate from donor mice Values are mean ± SEM with corresponding data points overlaid. Error bars are SEM. Source data are provided as a Source Data file. Scale bars are 25 μm. SRS—subretinal space, OPL—outer plexiform layer, INL—inner nuclear layer, GCL—ganglion cell layer, P—postnatal day

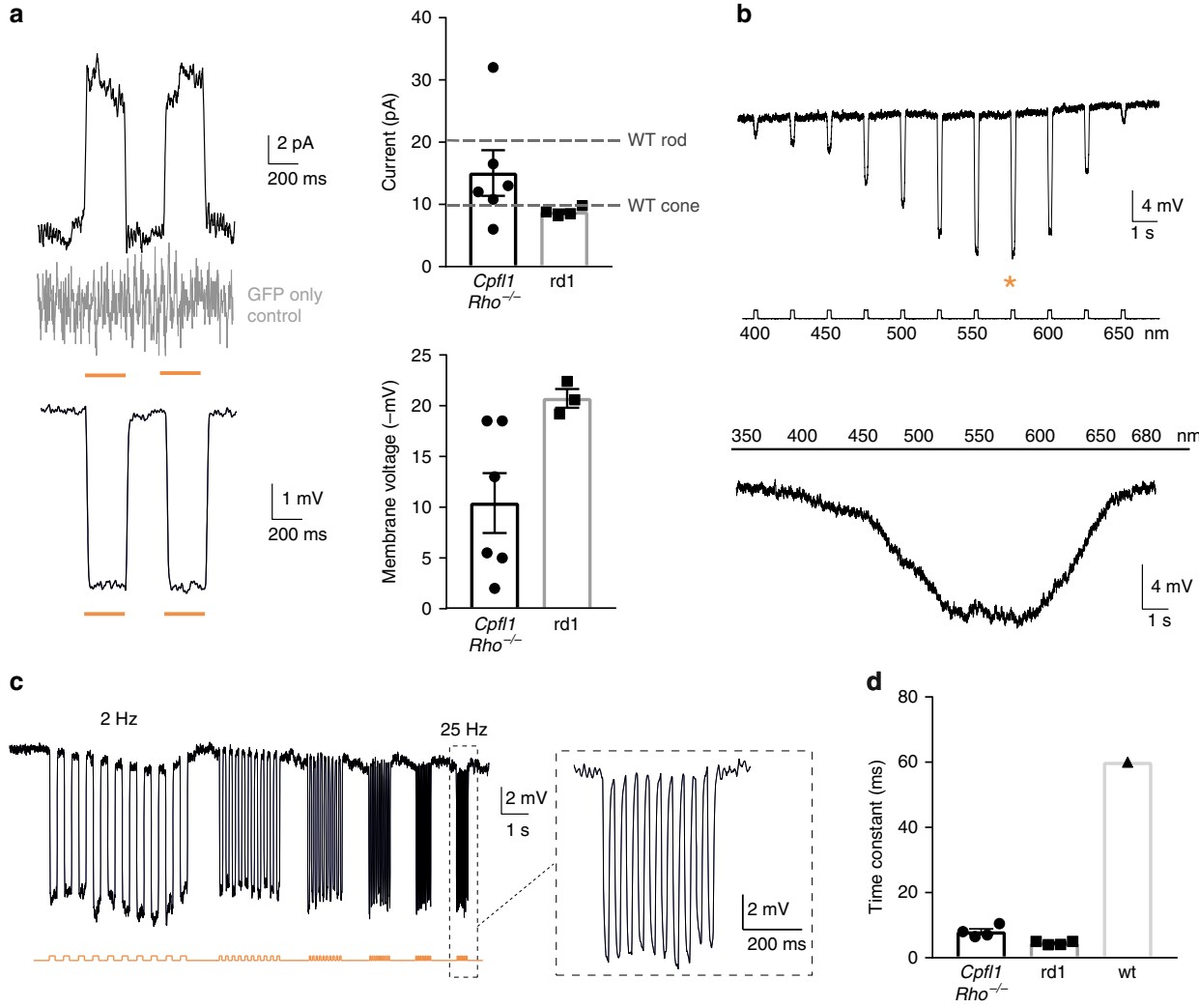

**Fig. 2** Transplanted NpHR-expressing photoreceptor precursors respond to light. Light response characteristics from cells recorded by whole-cell patch-clamp technique in treated $Cpfl1/Rho^{-/-}$ mice. The resting membrane potential (RMP) of transplanted photoreceptors in the dark (at 0 current) for the recordings presented in the figure was −36 ± 1.5 mV. **a** Left, light-evoked responses of NpHR- photoreceptors stimulated with two consecutive flashes (top, current response; bottom, voltage response), absence of the response in GFP only expressing photoreceptor shown in grey. Right, comparison of response amplitudes. Mean photocurrent peak (top) and mean peak voltage response (bottom). Mean values observed in wild-type rods and cones are indicated with a dashed line[58]. **b** Representative action spectrum from a NpHR photoreceptor stimulated at different wavelengths. Top, stimuli ranging from 400 nm to 650 nm, separated by 25 nm steps. Maximal voltage responses were obtained at 575 nm (denoted with an orange star). Bottom, continuous 'rainbow' stimulation between 350 and 680 nm. **c** Temporal properties: Modulation of NpHR-induced voltage responses at increasing stimulation frequencies from 2 to 25 Hz. **d** Comparison of rise time constants in the two models and in wild-type cones. In all panels: Light stimulations were performed at $8.7 \times 10^{16}$ photons cm$^{-2}$ s$^{-1}$ and 590 nm, if not stated otherwise. The timing and duration of stimulation is depicted with underlying orange lines (for 590 nm stimuli; **a, c**) or with a black line with associated wavelengths noted below (**b**). $n =$ number of cells. Values are mean ± SEM with corresponding data points overlaid. Error bars are SEM. Source data are provided as a Source Data file (for **a, d** and RMP)

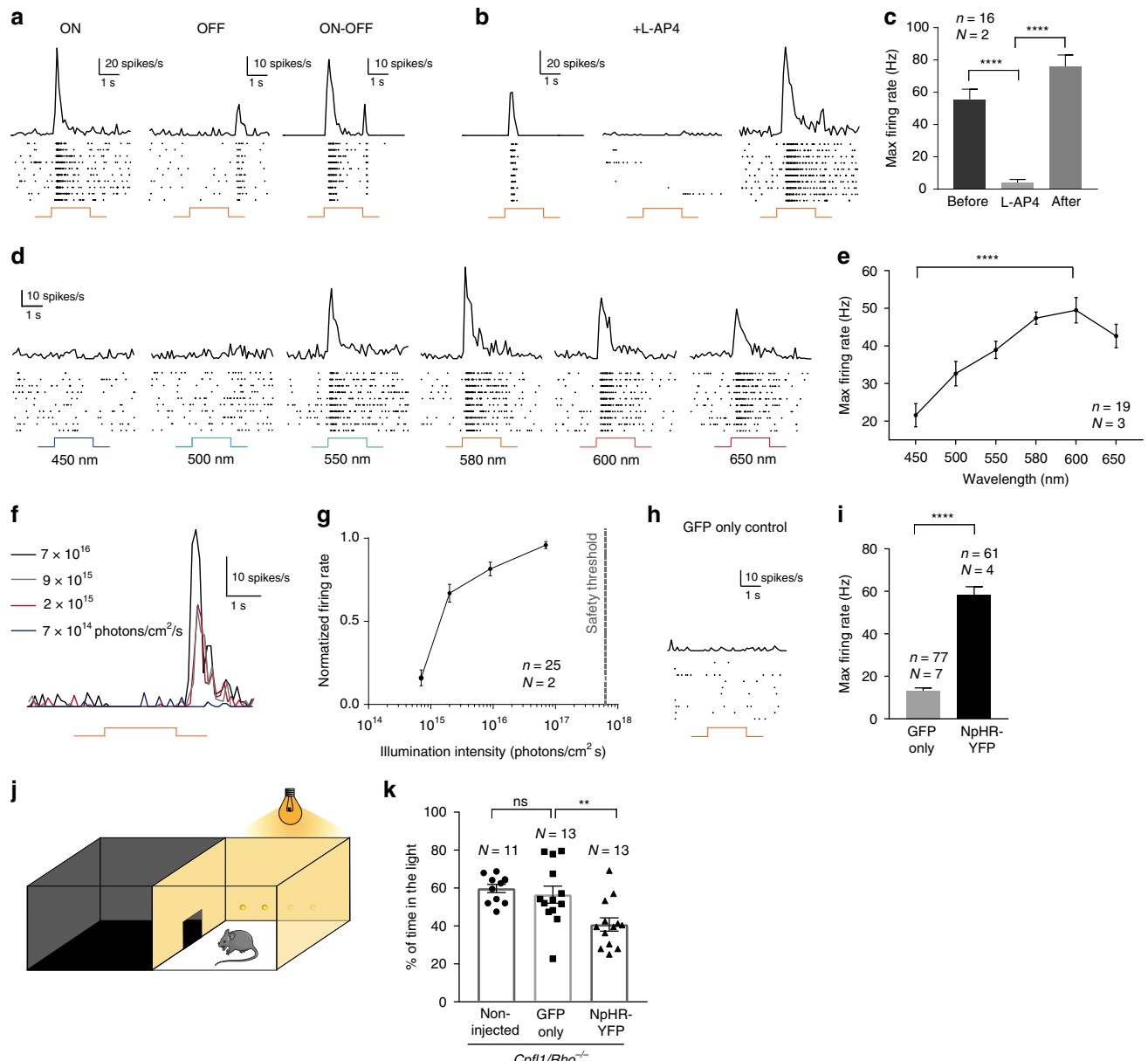

**Fig. 3** NpHR-triggered responses are transmitted to RGCs and induce light avoidance behaviour. **a–i** Averaged spike responses obtained from multi-electrode array (MEA) recordings shown as PSTH and raster plots recorded in transplanted $Cpfl1/Rho^{-/-}$ mice (stimulation: 580 nm, $7 \times 10^{16}$ photons cm$^{-2}$ s$^{-1}$). **a** Representative traces from three RGCs responding either with an ON-, OFF-, or ON/OFF-response pattern. **b** Representative traces from a cell before, during ON bipolar cell blockade, and after wash-out, and **c** quantification of maximum firing rates for these conditions. **d** Representative responses to wavelengths ranging from 450 nm to 650 nm. **e** Quantification of RGC action spectrum (shown for OFF responses). The cells reach their peak firing rate at 580 nm (ON responses, data not shown) and 600 nm (OFF responses). **f** PSTHs of a single RGC responding to stimuli of increasing intensities (from $7 \times 10^{14}$ to $7 \times 10^{16}$ photons cm$^{-2}$ s$^{-1}$). **g** Intensity curve. The dashed line indicates the maximum light intensity allowed in the human eye at 590 nm[20, 21]. **h** Unresponsive cell from a control retina transplanted with GFP only-expressing photoreceptors. **i** Maximum firing rate in mice treated with GFP only photoreceptors versus mice treated with NpHR-photoreceptors (shown for ON responses). **j** Schematic representation of the dark/light box test. **k** Percentage of time spent in the light compartment for: non-treated $Cpfl1/Rho^{-/-}$ mice, $Cpfl1/Rho^{-/-}$ mice treated with GFP only photoreceptors, and $Cpfl1/Rho^{-/-}$ mice treated with NpHR-photoreceptors (illumination: 590 nm, $2.11 \times 10^{15}$ photons cm$^{-2}$ s$^{-1}$). In all panels: The timing and duration of stimulation is depicted with underlying orange lines (for 580 nm stimuli; **a**, **b**, **f**, **h**) or with a black line with associated wavelengths noted below (**d**). $N$ = number of retinas, $n$ = number of cells. Values are mean ± SEM. Corresponding data points are overlaid in (**k**). Error bars are SEM. Statistical significance assessed using Mann–Whitney Student's test (**$p < 0.01$; ****$p < 0.0001$; ns—not significant). Source data are provided as a Source Data file (for **c**, **e**, **g**, **i**, **k**).

encoding Jaws-GFP under the control of CAR promoter, we delivered the microbial opsin to the hiPSC-derived cone photoreceptors (Fig. 4g, h). Single cell recordings from optogenetically transformed cones in retinal organoids revealed solid light responses, matching the response properties of Jaws, while

recordings from hiPSC-derived cones, expressing GFP only, showed no light responses (Fig. 4i–l). Additionally, monolayer cultures of these human cones expressing Jaws, maintained their ability to strongly respond to light after dissociation of the retinal organoids (Supplementary Fig. 7). These results collectively

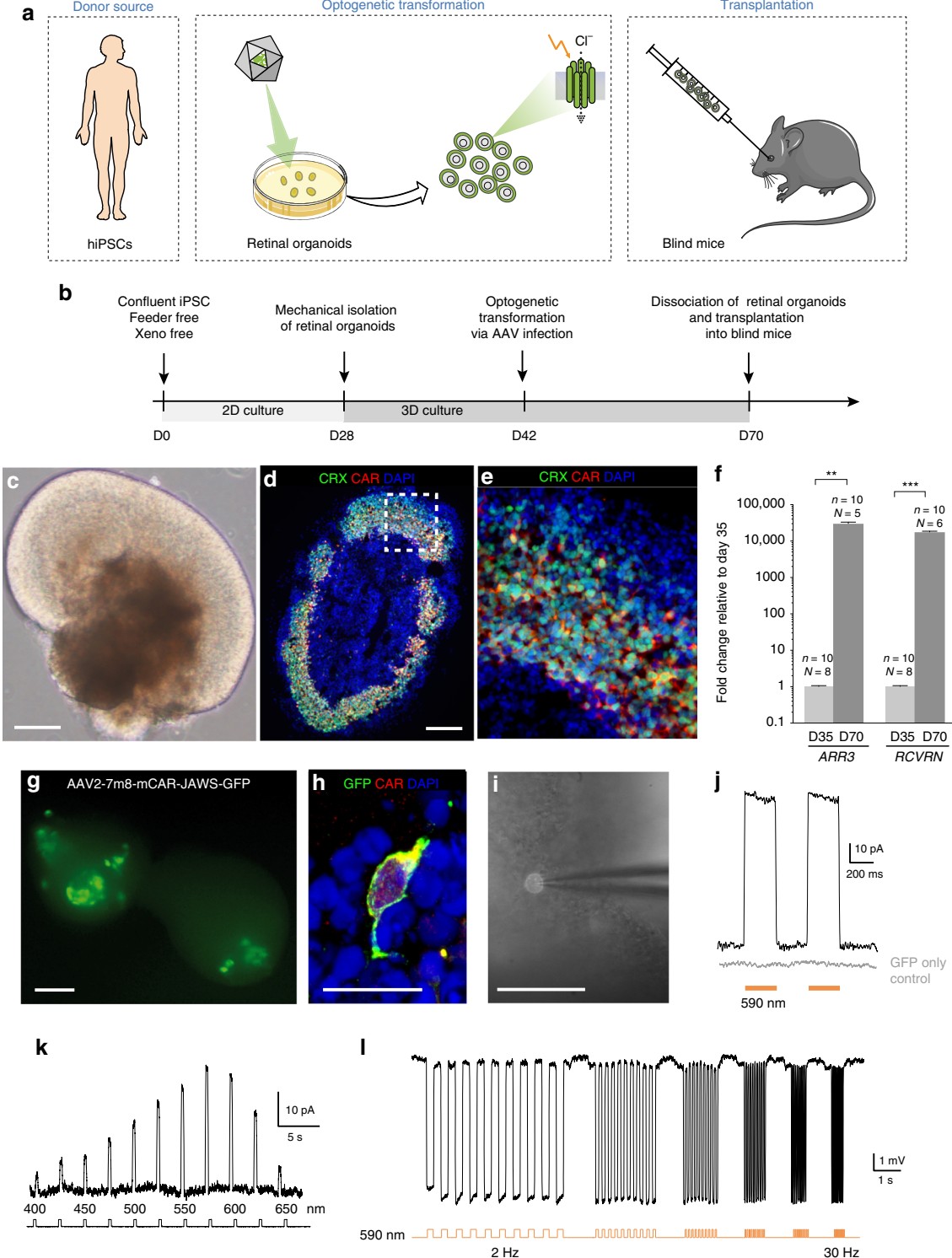

demonstrate the possibility to induce robust optogenetic light responses in photoreceptors derived from hiPSCs in the absence of light sensitive OS.

**Transplantation of hiPSC-derived Jaws-expressing PRs.** In order to transplant Jaws-positive photoreceptors, we dissociated the retinal organoids and injected the cell suspension subretinally into the blind hosts ($Cpfl1/Rho^{-/-}$, age 10 to 15 weeks; rd1, age 4 to 5 weeks). In both $Cpfl1/Rho^{-/-}$ and rd1 mice, we observed

Jaws-expressing donor cells in close proximity to the host INL several weeks after transplantation (Fig. 5a–c). Due to recent concerns about material transfer in photoreceptor transplantation[27–29], we stained cryosections from the transplanted retinas with the human nuclear antigen (HNA) and we examined the size of the transplanted cells ($HNA^+$) in relation to the chromatin structure and diameter of host cells (Fig. 5d). HNA stained cell counts confirmed that only a very small portion (5%) of the $GFP^+$ labeled cells could potentially be endogenous mouse cells that underwent material transfer ($HNA^-/GFP^+$) (Supplementary

**Fig. 4** Jaws-expressing photoreceptors, derived from hiPSCs, are sensitive to light. **a** Human iPSCs were differentiated towards retinal organoids and were infected with AAV-mCar-Jaws-GFP. After further maturation, cells were dissociated and iPSC-derived photoreceptors were transplanted into blind mice. **b** Schematic diagram of the differentiation and viral transformation of retinal organoids. **c** Bright-field image of a retinal organoid at D30 of differentiation. **d, e** Characterization of a representative retinal organoid at D70, depicting a thick layer of photoreceptors immunoreactive for CRX (green) and CAR (red). **f** Real-time qRT-PCR analysis of photoreceptor specific markers *CAR (ARR3)* and *RCVRN*. N = number of biological replicates, *n* = number of organoids. Values are mean ± SEM. Error bars are SEM. Statistical significance assessed using Mann–Whitney Student's test (**$p < 0,01$; ***$p < 0,001$). Source data are provided as a Source Data file. **g** Live GFP fluorescence observed at D54 (12 days post infection). **h** A single cone photoreceptor stained with GFP (green) and CAR (red) at D70. **i** Bright field/epifluorescence image of a GFP$^+$ cell patched inside a retinal organoid at D70 of differentiation. Scale bars are 100 μm (**c, d, g, i**) and 25 μm (**e, h**). **j–l** Patch-clamp data from Jaws-cones within organoids. The resting membrane potential (RMP) of Jaws-expressing photoreceptors in the dark (at 0 current) for the recordings presented in the figure was -41,7 ± 3,9 mV. Source data are provided as a Source Data file. Stimulation at 590 nm if not stated otherwise. **j** Photocurrent responses after stimulation with two consecutive flashes at $3.5 \times 10^{17}$ photons cm$^{-2}$ s$^{-1}$, absence of response in GFP only expressing cones is shown in grey. **k** Photocurrent action spectrum corresponding to a Jaws-cone stimulated at wavelengths ranging from 400 to 650 nm. Maximal responses were obtained at 575 nm (at $8.7 \times 10^{16}$ photons cm$^{-2}$ s$^{-1}$). **l** Modulation of Jaws-induced voltage responses at increasing stimulation frequencies from 2 to 30 Hz. The timing and duration of stimulation is depicted with underlying orange lines (for 590 nm stimuli; **j, l**) or with a black line with associated wavelengths noted below (**k**)

Fig. 8). Both the HNA staining and nuclei comparison confirmed the human identity of transplanted cells in close proximity of the host INL. The transplanted GFP$^+$ cells were RCVRN positive (Supplementary Fig. 8) and located next to PKCα-positive bipolar cells (Fig. 5a). They expressed the synaptic marker Synaptophysin in close apposition to the bipolar cell dendrites (Fig. 5b), suggesting that the human cells form synaptic connections with the host bipolar cells. The transplanted cells displayed robust Jaws-induced photocurrents by patch clamp, demonstrating the functionality of the microbial opsin in the host environment (Fig. 5e). The measured photocurrents peaked at 575 nm and showed fast kinetics (Tau$_{ON}$ < 10 ms) (Fig. 5f–h), reflecting the response properties of Jaws. At the ganglion cell level, we observed ON- and OFF responses from different ganglion cell types, which shows that Jaws-driven signals from transplanted photoreceptors were transmitted via second order neurons (Supplementary Fig. 9) to ON and OFF ganglion cells (Fig. 5i and Supplementary Fig. 10). The light intensity requirements were again below the safety threshold for the human retina[20,21] (Fig. 5j and Supplementary Fig. 10). After transplantation of control human donor cells, expressing GFP only, no light responses were detected (Fig. 4k and Supplementary Fig. 10), as expected in absence of OS-like structures.

## Discussion

Transplantation of healthy photoreceptors holds great promise to restore vision in patients with outer retinal degeneration. This approach has received significant attention over the past years as it can restore vision independently from the cause of photoreceptor cell loss[30]. Significant progress has been made in the generation[23,26,31–35], purification[36,37] and transplantation of photoreceptors[3,7,35,36,38–40] from hiPSCs. However, photoreceptor replacement faces a three-fold challenge: transplanted cells need to develop (1) synaptic contact to bipolar cells for signal transmission, (2) functional photoreceptor OS, and (3) tight contact to RPE cells to maintain OS light-sensitivity (Fig. 6). This makes photoreceptor transplantation complex and challenging. Recent studies have shown that the recipient environment is of great importance for successful integration and survival of transplanted photoreceptor cells. In animals with severely degenerated ONL, transplanted photoreceptor precursors derived from postnatal mouse retina[4–6] or from hiPSCs[35] failed to develop normal OS structure and establish correct OS polarity with respect to host RPE. The RPE cells are indispensable for OS renewal as they phagocytose the shed OS discs. Moreover, they re-isomerize the chromophore all-*trans*-retinal into 11-*cis*-retinal. Thus, in the absence of intimate contact with the RPE photoreceptors cannot maintain their light sensitivity[41].

For an OS and RPE-independent treatment approach, we introduced a hyperpolarizing microbial opsin into photoreceptors derived from either neo-natal mouse retinas or from human retinal organoids derived from iPSCs. We transplanted these optogenetically-transformed photoreceptors into blind mice lacking the photoreceptor layer. We have shown that these cells can mediate visual function, as demonstrated by a battery of tests from RGC recordings to behavioural tests. The paradigm that transplanted photoreceptors migrate and structurally integrate into the ONL of the recipient has been challenged recently by several groups[27–29] providing strong evidence that cytoplasmic material transfer occurs between transplanted cells, residing in the subretinal space, and remaining photoreceptor cells of the host. In these experiments, however, late-stage degeneration animals were used to model patients with advanced disease, thus there are only few remaining photoreceptors, minimizing the potential contribution of material transfer[42]. To distinguish between potential fusion events and structural integration of donor photoreceptors, we performed Y chromosome FISH and HNA staining in the *Cpfl1/Rho*$^{-/-}$ model where some remaining cells were visible in earlier transplantation time-points. Our Y chromosome FISH experiments revealed a very limited number of events of potential material transfer (<10%). In our blind rd1 mice, only sparse population of cones and no rods remain after 36 days of age[18]. We confirmed this observation in our control animals, obviating the possibility of material transfer from the transplanted NpHR-expressing mouse progenitors to remnant ONL cells of the host. Moreover, NpHR-positive cells that were attached to the host INL visibly show rod nuclear morphology, indicating that these are indeed donor cells and not remaining cones. As for the transplantation of Jaws-expressing hiPSCs, histological analysis using a human-specific nuclear marker (HNA) in transplanted mice, confirmed that the vast majority (95%) of GFP-expressing cells were HNA positive. This result along with the measured nuclei size confirmed the human origin of the transplanted cells, ruling out material exchange between human donor photoreceptors and mouse host cells. Although these do not fully rule out that material transfer may contribute to the improved functional responses, we have observed that the level of functional improvement is independent of the host age at time of transplantation, further supporting the optogenetically transformed photoreceptors are the major source of functional light responses (Supplementary Fig. 11). Moreover, material transfer is rare between human donor and mouse host photoreceptors[35,36] (Fig. 5d and Supplementary Fig. 8), arguing against a significant contribution of material transfer to the observed functional improvements.

Lastly, any possible rescue effect mediated by remaining host photoreceptors is expected to be very minor as our control groups

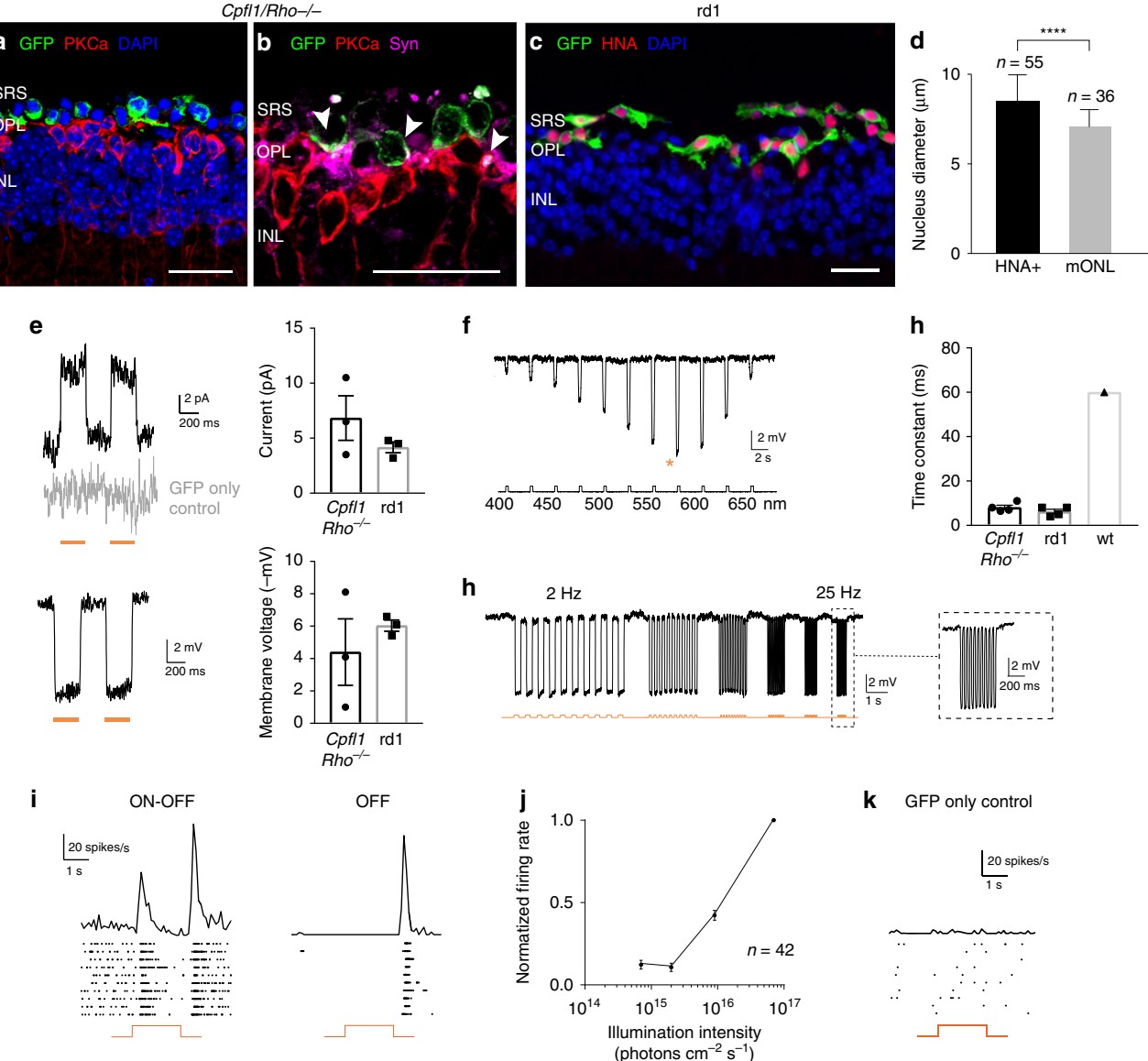

**Fig. 5** hiPSC-derived photoreceptors display Jaws-induced light responses that are transmitted to RGCs. Immunofluorescence analysis of Jaws-cone-treated *Cpfl1/Rho*$^{-/-}$ (**a**, **b**) and rd1 (**c**) retinas. **a** Transplanted cells (green) overlie host PKCα bipolar cells (red), DAPI counterstaining (blue). **b** GFP, PKCα and synaptophysin staining. Arrows point to synaptic connections. **c** GFP$^+$ Jaws-cones co-express HNA. Scale bars are 20 μm. SRS—subretinal space, OPL—outer plexiform layer, INL—inner nuclear layer. **d** Measurement of nuclear size of HNA$^+$ cells, transplanted in rd1 mice, and ONL cells of a wild-type mouse. **e–h** Patch-clamp data from Jaws-cones after transplantation into blind mice. The RMP of Jaws-photoreceptors at 0 current was −40.8 ± 5.2 mV. Stimulation at 590 nm if not stated otherwise. **e** Left, representative photocurrents (top) and voltage hyperpolarization (bottom) after stimulation with two consecutive flashes, absence of the response in GFP only cones shown in grey. Right, comparison of response amplitudes of Jaws-cones in different models (top, mean photocurrent peak; bottom, mean voltage peak). **f** Voltage action spectrum corresponding to a Jaws-expressing cell stimulated at wavelengths from 400 to 650 nm. Maximal responses were obtained at 575 nm (orange star). **g** Temporal properties: Jaws-induced hyperpolarization at increasing stimulation frequencies from 2 to 25 Hz. **h** Comparison of response rise time constant between Jaws-cones transplanted in *Cpfl1/Rho*$^{-/-}$ and rd1 models, and wild-type cones. **i–k** Averaged spike responses obtained from MEA recordings shown as PSTH and raster plots from a transplanted *Cpfl1/Rho*$^{-/-}$ mouse. **i** Representative examples of ON/OFF and OFF-responding RGCs (stimulation: 580 nm, 7 × 10$^{16}$ photons cm$^{-2}$ s$^{-1}$). **j** Intensity curve. **k** Recording from a control retina transplanted with GFP only cones. In all panels: Stimulations are depicted with underlying orange lines (for 580–590 nm stimuli; **e**, **g**, **i**, **k**) or with a black line with associated wavelengths noted below (**f**). *n* = number of cells. Values are mean ± SEM. Corresponding data points are overlaid in (**e**, **h**). Error bars are SEM. Statistical significance assessed using Mann–Whitney Student's test (****$p$ < 0,0001). Source data are provided as a Source Data file (for **d**, **e**, **h**, **j** and RMP)

transplanted at the same ages with wild-type donor-derived photoreceptor precursors or hiPSC-derived photoreceptors expressing GFP only, never showed any detectable functional responses. This confirms that any possible rescue effect on remaining host photoreceptors cannot be a result of the transplantation itself and suggests that the functional outcomes are a direct consequence of the presence of an optogenetic protein expressed in the transplanted photoreceptors.

In conclusion, by using immature photoreceptors equipped with a microbial opsin, we went beyond the current limitations of optogenetic gene therapy approaches. Optogenetic approaches commonly target bipolar cells or RGCs that are viable targets in late

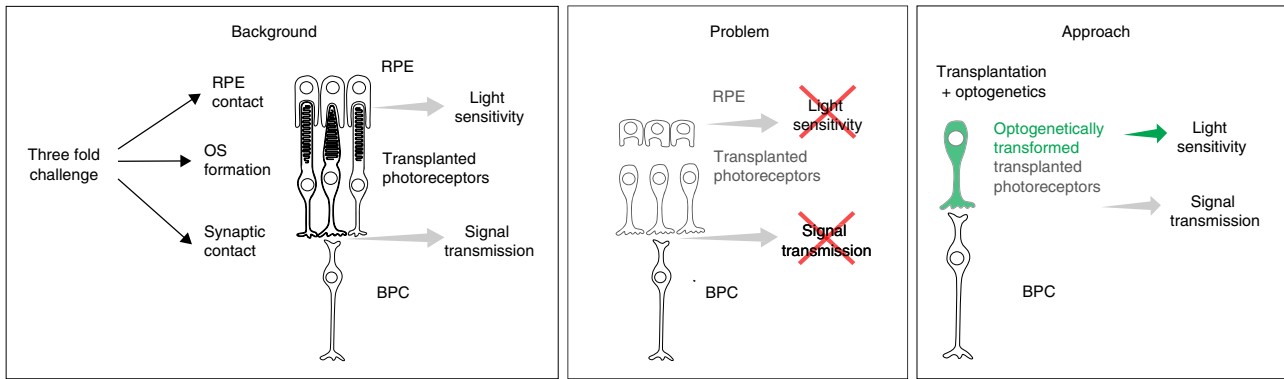

**Fig. 6** Schematic illustrating the three-fold challenge in photoreceptor cell replacement. In order to provide visual improvement, transplanted photoreceptors need to form functional OS, retain in close contact to the RPE to maintain light sensitivity, and develop synaptic connection to host bipolar cells for signal transmission. After transplantation into animals with severely degenerated ONL, photoreceptors fail to develop normal OS structure and establish correct polarity with respect to host RPE. In addition, in retinal degeneration, the RPE is often compromised alongside photoreceptors. All this undermines the success of photoreceptor replacement. We therefore introduced a hyperpolarizing microbial opsin into the photoreceptors before transplantation, developing an OS- and RPE-independent approach for vision restoration in late-stage retinal degeneration. RPE—retinal pigment epithelium, PR—photoreceptors, BPC—bipolar cells

stages of retinal degenerative diseases such as retinitis pigmentosa or age-related macular degeneration. Unfortunately, conferring light sensitivity to cells downstream from photoreceptors, bypasses the important information processing normally conducted by the inner retinal circuitry. Photoreceptor-directed optogenetic therapy that aims to rescue the function of remaining dormant cones harnesses the information processing of the inner retina allowing the recovery of complex visual responses such as lateral inhibition and directional selectivity in previously blind mice[43], but this strategy can only be useful in patients with remaining cones which represent a minor portion of late-stage retinitis pigmentosa patients[44]. Here, we use the synergy of cell replacement and optogenetic therapy that allows the restoration of retinal structure with stem cell derivatives and visual function with microbial opsins. In a future perspective, optogenetically engineered hiPSC-derived cones could serve as donor cells for photoreceptor transplantation in late-stage retinal degeneration. In patients, degenerative diseases of the retina such as retinitis pigmentosa, age-related macular degeneration, and Leber congenital amaurosis, often manifest RPE degeneration along with photoreceptor degeneration, especially in their late stages[10,45–47]. Our approach bodes well for applications in such patients who can only obtain limited benefit from transplantation of photoreceptors in the absence of chromophore replenishment from their dystrophic RPE.

## Methods

**Animals**. Wild-type C57BL/6 mice (Janvier Laboratories) were used as a source of photoreceptor precursor donor cells. The following two models are both models of late-stage degeneration and were used as cell recipients. Cone photoreceptor function loss 1/rhodopsin-deficient double-mutant $Cpfl1/Rho^{-/-}$ mice[16] were provided by Marius Ader and rederived by Charles River Laboratory. The line was the result of crossing Cone photoreceptor function loss 1 ($Cpfl1$) mice[48] with rhodopsin knock-out mice ($Rho^{-/-}$)[49]. The outcome were mice with no functional photoreceptors starting from eye opening and with the ONL degenerating to one row of cell bodies by 10 to 12 weeks[16]. Retinal degeneration 1 (rd1) mice ($C3Hrd/rd$)[17] were provided by Thierry Leveillard. The retina in these mice degenerates to a single row of cones by 3 weeks of age[18,50].

All mice were housed under a 12-h light-dark cycle with free access to food and water. We have complied with all relevant ethical regulations for animal testing and research. All animal experiments and procedures were approved by the local animal experimentation ethics committee (Le Comité d'Ethique pour l'Expérimentation Animale Charles Darwin) and were carried out according to institutional guidelines in adherence with the National Institutes of Health guide for the care and use of laboratory animals as well as the Directive 2010/63/EU of the European Parliament.

**AAV production**. Recombinant AAVs were produced using the triple-transfection method on HEK293 cells (ATCC CRL-1573), harvested 24–72 h post transfection

and purified by iodixanol gradient ultracentrifugation[51]. The 40% iodixanol fraction was collected after a 90 min spin at 354000 g. Concentration and buffer exchange were performed against PBS containing 0.001% Pluronic. AAV vector stocks titers were then determined based on real-time quantitative PCR titration method using ITR primers[52] and SYBR Green (Thermo Fischer Scientific).

**AAV-infection of photoreceptor precursors**. Wild-type mice (C57BL/6J) at P2 were anesthetized on ice. Eyelids were cut and 1 µl of AAV9 2YF carrying eNpHR gene under the control of human rhodopsin promoter and fused to the fluorescent reporter eYFP (AAV9 2YF hRho-eNpHR-eYFP), or of AAV9 2YF hRho-GFP in the case of GFP only-expressing controls, was injected bilaterally using an ultrafine 34-gauge Hamilton syringe.

**MACS with CD73**. Two days following the AAV injections in P2 mice, at P4, retinas were isolated from the injected wild-type mice and cells were enriched using CD73 cell surface marker before transplantation[14,15]. Briefly, retinas were dissociated, pelleted by centrifugation (5 min at 300 g), resuspended in 500 µL MACS buffer (phosphate-buffered saline [PBS; pH 7.2], 0.5% BSA, 2 mM EDTA) and incubated with 10 µg/mL rat anti-mouse CD73 antibody (BD Biosciences, 550738) for 5 minutes at 4 °C. After washing in MACS buffer, cells were centrifuged for 5 minutes at 300 g. The cell pellet was resuspended in 480 µL MACS buffer and 120 µL goat anti-rat IgG magnetic beads (Miltenyi Biotec, 130-048-501). The suspension was incubated for 15 min at 4 °C followed by a washing step with MACS buffer and centrifugation. Before magnetic separation, the cells were resuspended in MACS buffer and filtered through a 30-µm pre-separation filter.

The cell suspensions were applied onto a LS column fixed to a MACS separator. The column was rinsed with 3 × 3 mL MACS buffer and the flow through was collected (CD73 negative cells). The column was removed from the magnet and placed in a new collection tube. The CD73-positive fraction was eluted by loading 5 mL MACS buffer and immediately applying the plunger supplied with the column. The cells were then counted and concentrated to about 200,000 cells/µl.

**Maintenance of hiPSC culture**. All experiments were carried out using hiPSC-2 cell line, previously established from human dermal fibroblasts from an 8-year-old boy (gift from P. Rustin, INSERM U676, Paris) by co-transfecting OriP/EBNA1-based epi-somal vectors pEP4EO2SEN2K (3 µg), pEP4EO2SET2K (3 µg) and pCEP4-M2L (2 µg) (Addgene) via nucleofection (Nucleofector 4D, V4XP, withDT-130 program; Lonza)[31], and recently adapted to feeder-free conditions[23]. Cells were kept at 37 °C, under 5% $CO_2$/95% air atmosphere, and 20% Oxygen tension and 80-85% of humidity. Colonies were cultured with Essential 8™ medium (Thermo Fisher Scientific) in culture dishes coated with truncated recombinant human Vitronectin and passaged once a week[23].

**Generation of retinal organoids from human iPS cells**. Human iPSC were differentiated towards retinal organoids following an optimized protocol based on the one published by Reichman et al.[23]. Briefly, hiPSC-2 cell line was expanded to 80% confluence in Essential 8™ medium were switched in Essential 6™ medium (Thermo Fisher Scientific). After 3 days, cells were moved to the Proneural medium (Supplementary Table 2). The medium was changed every 2–3 days. After 4 weeks of differentiation, neural retina-like structures grew out of the cultures and were mechanically isolated. Pigmented parts, giving rise to RPE were carefully removed. The extended 3D culture in Maturation medium (Table S1) allowed the

formation of retinal organoids. Addition of 10 ng/ml Fibroblast growth factor 2 (FGF2, Preprotech) at this point favoured the growth of retinal organoids and the commitment towards retinal neurons instead of RPE lineage[53]. In order to promote the commitment of retinal progenitors towards photoreceptors, we specifically blocked Notch signalling for a week starting at day 42 of differentiation using the gamma secretase inhibitor DAPT (10 μM, Selleckchem)[54]. Floating organoids were cultured in 6 well-plates (10 organoids per well) and medium was changed every 2 days. Supplementary Table 2 summarizes the formulations for the different media used.

**Infection of retinal organoids with AAV expressing Jaws**. Introduction of Jaws optogene was done by one single infection at day 42 at a $5 \times 10^{10}$ vg per organoid. Retinal organoids were infected with an AAV with an engineered capsid, AAV2-7m8[55] carrying Jaws gene under the control of mouse cone arrestin promoter and fused to the fluorescent reporter GFP (AAV2-7m8-mCAR-Jaws-GFP). For GFP only-expressing controls, an infection with AAV2-7m8-mCAR-GFP was carried out in the same manner as mentioned above.

**Monolayer cultures of dissociated cells**. After removal of any pigmented tissue, 70-day old retinal organoids were collected and washed three times in Ringer solution (Supplementary Table 2) before dissociation with two units of pre-activated papain at 28.7 μ/mg (Worthington) in Ringer solution for 25 min at 37 °C. Once a homogeneous cell suspension was obtained after pipetting up and down, papain was deactivated with Proneural medium (Supplementary Table 2). Cells were centrifuged and resuspended in pre-warmed Proneural medium. Dissociated retinal cells were plated onto coverslips coated with human recombinant 30 μg/cm² Laminin (Sigma-Aldrich) and 150 μg/cm² Poly-L-Ornithine in 24 well-plates[56]. Monolayers were incubated at 37 °C in a standard 5% CO2/95% air incubator and medium was changed every 2 days for the next 15–20 days, before immunostaining.

**Preparation of cells for transplantation**. At day 70 of differentiation retinal organoids were dissociated using papain as described above to obtain a single cell suspension in Proneural medium (Supplementary Table 2). Cell suspension was filtered through a 30 μm mesh (Miltenyi Biotec) to remove residual aggregates. After counting, cells were centrifuged and resuspended in Proneural medium at a concentration of 300,000 cells/μl.

**RNA isolation and real-time RT-qPCR**. Total RNA isolation was performed using a NucleoSpin RNA XS kit (Macherey-Nagel), according to the manufacturer's instructions. RNA concentration and purity were determined using a NanoDrop ND-1000 Spectrophotometer (Thermo Fisher Scientific).

Reverse transcription was carried out with 250 ng of total RNA using the QuantiTect retrotranscription kit (Qiagen). Quantitative PCR (qPCR) reactions were performed using Taqman Array Fast plates and Taqman Gene expression master mix (Thermo Fisher Scientific) in an Applied Biosystems real-time PCR machine (7500 Fast System). All samples were normalized against a housekeeping gene (18S) and the gene expression was determined based on the ΔΔCT method. Average values were obtained from at least four biological replicates. The primer sets and MGB probes (Thermo Fisher Scientific) labelled with FAM for amplification are listed in Supplementary Table 3.

**Transplantation procedure**. Mice were sedated by intraperitoneal injection of ketamine (50 mg/kg) and xyazine (10 mg/kg) and the pupils were dilated with tropicamide drops. The mice were placed onto a heating pad to maintain the temperature at 37 °C. A drop of Lubrithal eye gel (Dechra) was used to keep the eyes hydrated during the surgery. A small glass slip was put on the eye to enable visualization through the Leica Alcon ophthalmic microscope while a syringe with a blunt, 34-gauge needle was inserted tangentially through the conjunctiva and sclera. One microlitre of cell suspension including 200,000–300,000 cells was injected between the retina and RPE, into the subretinal space, creating a bullous retinal detachment. Injections were performed bilaterally. Mice were placed into a warm chamber after the surgery until their awakening.

**Tissue preparation and immunostaining**. 70-day old organoids were washed in PBS and fixed in 4% paraformaldehyde for 10 min at 4 °C before they were incubated overnight in 30% sucrose (Sigma-Aldrich) in PBS. Organoids were embedded in gelatin blocks (7.5% gelatin (Sigma-Aldrich), 10% sucrose in PBS) and frozen using isopentane at −50 °C.

At least 4 weeks after transplantation, mice were sacrificed by CO2 inhalation followed by a cervical dislocation. The eyeballs were removed, fixed in 4% paraformaldehyde for 30 minutes at room temperature (RT) and incubated overnight at 4 °C in PBS containing 30% (w/v) sucrose (Sigma-Aldrich). The eyes were then dissected to obtain only the back of the eye with the retina and the RPE. The samples were embedded in gelatin blocks (7.5% gelatin (Sigma-Aldrich), 10% sucrose in PBS), frozen with liquid nitrogen and stored at −80 °C.

Ten micrometres of thick sections were obtained using a Cryostat Microm and mounted on Super Frost Ultra Plus® slides (Menzel Gläser). Cryosections were washed in PBS (5 min, RT) and then permeabilised in PBS containing 0.5 % Triton

X-100 during 1 h at RT. Blocking was done with PBS containing 0.2% gelatin, 0.25% Triton X-100 for 30 min at RT and incubation with primary antibodies was performed overnight at 4 °C. We used the following primary antibodies for immunostaining: hCAR (1:20,000; gift from Cheryl Craft), CRX (1:5000; Abnova, H00001406-M02), GFP (1:500; Abcam, ab13970), HNA (1:200; Millipore, MAB4383), Ki67 (1:200; BD Pharmgen, 550609), PKCα (1:100; Santa Cruz, sc-208), RCVRN (1:5000-1:2000; Millipore, AB5585), Synaptophysin (1:200; Sigma, SAB4502906). The antibodies used are also listed in Supplementary Table 4. After incubation with primary antibodies, sections were washed with PBS containing 0.25% Tween20 and incubated with fluorochrome-conjugated secondary antibodies (1:500; Thermo Fisher Scientific) for 1 h at RT. After successive washing in PBS-Tween20, nuclei were counterstained with DAPI (4′-6′-diamino-2-phenylindole, dilactate; Invitrogen-Molecular Probe, Eugene, OR) at a 1:1000 dilution. Samples were further washed in PBS and dehydrated with 100% ethanol before mounting using fluoromount Vectashield (Vector Laboratories).

**FISH for Y chromosome detection**. For combined chromosomal fluorescence in situ hybridization (Y chromosome FISH) and immunohistochemistry, retinas from female $Cpfl1/Rho^{-/-}$ mice transplanted with male donor-derived rod precursors ($N = 5$) were collected 4 weeks post-surgery, fixed for 1 h at 4 °C with freshly prepared 4% paraformaldehyde (Merck Millipore), incubated in 30% sucrose overnight, followed by cryopreservation. After embedding and freezing in OCT medium, cryosections of 12 μm were rehydrated with 10 mM sodium citrate buffer pH 6, antigen retrieval performed (80 °C, 25 min). Sections were washed in PBS for 5 min and incubated with a primary antibody against GFP (1:500; Abcam, ab13970) overnight at RT, followed by incubation with secondary antibody conjugated to AlexaFluor 488 (1:1000; Jackson Immunoresearch, 103545155) overnight at RT. Next, slides were post-fixed in 2% PFA for 10 min, pre-treated with 50% formamide for 1 h at RT, then hybridization of the XMP Y orange probe (Metasystems, D-1421-050-OR) to the Y chromosome was performed. To allow the probe to penetrate the tissue, samples were incubated for 3 h at 45 °C in a HybEZ II oven. Then, samples were transferred to a hot block at 80 °C for 5 min, to denature DNA. Afterwards, probes were hybridized with DNA for 2 days at 37 °C. Posthybridization consisted of 3 × 15 min washes with 2× SCC at 37 °C and 2 × 5 min stringency washes with 0.1× SCC at 60 °C. Finally, sections were counterstained with DAPI (1:15,000; Sigma). The samples were imaged and quantified using structured illumination microscopy (SIM; ApoTome, Zeiss).

For information on antibodies used, see Supplementary Table 4.

**Quantification of YFP⁺ cells after transplantation**. Tranplanted host eyes ($N = 6$) were processed and cryosectioned as described for the Y chromosome FISH experiment, and subsequently stained for GFP (1:500; Abcam, ab13970) and photoreceptor specific marker RCVRN (1:5000; Millipore, AB5585), followed by secondary antibody staining (1:1000; Jackson Immunoresearch) Every fourth serial section from whole experimental retinas was used to quantify the total amount of YFP⁺ photoreceptors. Cells were counted from images obtained with the Nano-Zoomer microscope (Hamamatsu Photonics). Following these cell counts, the resulting value was multiplied by four to estimate the total amount of labelled cells per retina.

For information on antibodies used, see Supplementary Table 4.

**Nuclear size measurements**. Measurements of the nuclear size were performed with FIJI software (NIH) on immunostained sections of rd1 transplanted retinas and compared with the values in wild-type mice.

**Image acquisition**. Immunofluorescence was observed using a Leica DM6000 microscope (Leica microsystems) equipped with a CCD CoolSNAP-HQ camera (Roper Scientific) or using an inverted or upright laser scanning confocal microscope (FV1000, Olympus) with 405, 488, 515 and 635 nm pulsing lasers. The images were acquired sequentially with the step size optimized based on the Nyquist–Shannon theorem. The analysis was conducted in FIJI (NIH). Images were put into a stack, Z-sections were projected on a 2D plane using the MAX intensity setting in the software's Z-project feature, and the individual channels were merged.

Images of Y chromosome labelled retinas were acquired using SIM (ApoTome, Zeiss). Samples stained to perform quantification of surviving YFP⁺ photoreceptors were imaged with the NanoZoomer microscope (Hamamatsu Photonics).

**Light stimulation of NpHR-positive, Jaws-positive, and control cells**. Light-triggered responses were measured in donor cells before transplantation—in vivo in AAV-injected wild-type donor mice at P12 for NpHR+, and in retinal organoids and monolayer cultures from dissociated organoids for Jaws+ cells. In order to measure light responses we used a monochromatic light source (Polychrome V, TILL photonics). After patching the cells we first stimulated them with a pair of 590 nm full-field light pulses. Then the activity spectrum was measured by using light flashes ranging from 400 to 650 nm (separated by 25 nm steps). Finally we generated light pulses at different frequencies ranging between 2 and 30 Hz in order determine the temporal response properties of NpHR and Jaws in AAV-transduced cells. Stimulation and analysis were performed using custom-written

software in Matlab (Mathworks) and Labview (National Instruments). We used light intensities ranging between $1 \times 10^{16}$ and $3.2 \times 10^{17}$ photons $cm^{-2} s^{-1}$.

**Live two-photon imaging and patch-clamp recordings**. Donor mouse retina (P12), retinal organoids or monolayer cultures from dissociated organoids were placed in the recording chamber of the microscope at 36 °C in oxygenated (95% $O_2$/5% $CO_2$) Ames medium (Sigma-Aldrich) during the whole experiment. Transplanted mice were sacrificed by $CO_2$ inhalation followed by quick cervical dislocation, and eyeballs were removed. Retinae from $Cpfl1/Rho^{-/-}$ or rd1 mice were isolated in oxygenated (95% $O_2$/5% $CO_2$) Ames medium and whole mount retinas with ganglion cell side down were placed in the recording chamber of the microscope at 36 °C for the duration of the experiment for both live two-photon imaging and electrophysiology.

A custom-made two-photon microscope equipped with a 25x water immersion objective (XLPlanN-25 × -W-MP/NA1.05, Olympus) equipped with a pulsed femto-second laser (InSight™ DeepSee™ - Newport Corporation) were used for imaging and targeting AAV-transduced fluorescent photoreceptor cells (eYFP+ or GFP+ cells). Two-photon images were acquired using the excitation laser at a wavelength of 930 nm. Images were processed offline using ImageJ (NIH). A CCD camera (Hamamatsu Corp.) was also used to visualize the donor cells or the retina under infrared light.

For patch-clamp recordings, AAV-transduced fluorescent cells were targeted with a patch electrode under visual guidance using the reporter tag's fluorescence. Whole-cell recordings were obtained using the Axon Multiclamp 700B amplifier (Molecular Device Cellular Neurosciences). Patch electrodes were made from borosilicate glass (BF100-50-10, Sutter Instrument) pulled to 7–10 MΩ and filled with 115 mM K Gluconate, 10 mM KCl, 1 mM $MgCl_2$, 0.5 mM $CaCl_2$, 1.5 mM EGTA, 10 mM HEPES, and 4 mM ATP-Na2 (pH 7.2). Photocurrents were recorded while voltage-clamping cells at a potential of −40 mV. Some cells were also recorded in current-clamp (zero) configuration, hence allowing us to monitor the membrane potential during light stimulations.

A monochromatic light source (Polychrome V, TILL photonics) was used to stimulate cells during electrophysiological experiments and hence record photocurrents or changes in cells membrane potential. First, in order to measure the activity spectrum of NpHR and Jaws, we used 300 ms light flashes ranging from 650 to 400 nm (25 nm steps; interstimulus interval 1.5 s) at a constant light intensity of $1.2 \times 10^{16}$ photons $cm^{-2} s^{-1}$. Then this light source was used at a constant wavelength of 590 nm to generate light pulses at different frequencies (ranging from 2 to 30 Hz) in order determine the temporal response properties of optogenetic proteins used. Stimuli were generated using custom-written software in LabVIEW (National Instruments) and output light intensities were calibrated using a spectrophotometer (USB2000+, Ocean Optics).

**Multi-electrode array recordings and data analysis**. The mice were euthanized, the retinas isolated, cut each in two pieces and placed in Ames medium bubbled with 95% $O_2$ and 5% $CO_2$. Each piece was mounted separately on a cellulose membrane soaked overnight in poly-L-lysin and gently pressed against a 60-μm electrode spacing 252 channel multi-electrode array chip (256MEA60/10iR, Multi Channel Systems) with RGCs facing the electrodes. The piece remained perfused with oxygenated Ames medium at 34 °C throughout the experiment. Full field light stimuli were applied with a Polychrome V monochromator (TILL Photonics) driven by a STG2008 stimulus generator (Multichannel Systems) using custom written stimuli in MC_Stimulus II (MC_Stimulus II Version 3.4.4, Multichannel Systems).

The basic stimulus pattern applied was 10 repetitions of 2-s stimuli of 580 nm light (close to excitation maximum for NpHR and Jaws) and intensity of $7 \times 10^{16}$ photons $cm^{-2} s^{-1}$, with 10 s intervals. To assess temporal dynamics of responding cells, stimuli ranging from 1 ms to 2 s were played to the retina. Action spectrum of optogenetic protein-expressing cells was examined by playing sets of stimuli of different wavelengths (450 nm, 500 nm, 550 nm, 580 nm, 600 nm, 650 nm; 10 stimuli of 2 s with 10 s intervals for each wavelength). To determine sensitivity of responding cells, stimuli of lower intensities were also used ($1 \times 10^{14}$, $7 \times 10^{14}$, $2 \times 10^{15}$ and $9 \times 10^{15}$ photons $cm^{-2} s^{-1}$). During the experiments aiming to show that the light responses are really coming from the ONL, we perfused the tissue with L-AP4 (50 μM) for at least 20 min before the recordings in order to block input from photoreceptors to ON bipolar cells. This was followed by at least 15 min rinse with Ames medium and another set of light stimulation to observe whether the response returned.

Data were acquired using the MC_Rack software (MC_Rack v4.5, Multi Channel Systems). RGC responses were amplified and sampled at 20 kHz. Data was then filtered with a 200 Hz high pass filter and individual channels were spike sorted using template matching and cluster grouping based on principal component analysis of the waveforms in Spike2 software v.7 (Cambridge Electronic Design Ltd). The raster plots and peristimulus time histogram data (bin size of 10 ms) were constructed in MATLAB using custom scripts from spike-sorted channels and further processed in Adobe Illustrator CS4 (Adobe Systems) for presentation.

Maximum firing rate for each responding cell was measured in the 2 s after the onset (for ON-responding cells) or 2 s after the offset (for OFF-responding cells) of the stimulus. The number of cells and mice that were used for quantitative analysis are stated in Figure legends. Error bars were calculated over cells.

**Light/dark box**. For light-avoidance behaviour, we used a custom-made dark-light box[22,57] of dimensions 36 cm × 20 cm × 18 cm, divided longitudinally into two equal sized compartments with a non-transparent wall with a 7 cm × 5 cm hole in the middle. The light compartment was equipped with eight 590 nm LEDs (Cree XP-E, amber, Lumitronix) 3 cm from the bottom of the box. A light intensity of $2.05 \times 10^{16}$ photons $cm^{-2} s^{-1}$ was used for all the experiments. The mice were habituated in the dark for at least 2 h prior the testing. Each mouse was introduced into the light compartment and was left in the box for at least 5 min before the start of illumination. The lights were turned on when the mouse was in the light compartment and were left on for at least 5 min. The behaviour of the mice was recorded with a camera and subsequently analyzed manually by recording the times spent in each compartment after the start of illumination, and using the Smart Vision Tracking Software (Harvard Apparatus). The mouse's head was used to define the compartment it occupied.

**Statistical analyses**. Data was analyzed with GraphPad Prism and it was expressed as mean ± standard error of mean (SEM). Comparisons between values were analyzed using unpaired two-tailed non-parametric Mann––Whitney Student's test. A level of $p < 0.05$ was considered significant. The labels used were: $*p < 0.05$, $**p < 0.01$, $***p < 0.001$ and $****p < 0.0001$.

**Reporting summary**. Further information on research design is available in the Nature Research Reporting Summary linked to this article.

## Data availability
The data that support the findings of this study are available from the corresponding author upon reasonable request. The source data underlying Figs. 1i, 2a, d, 3c, e, g, i, k, 4f, 5d, e, h, j, resting membrane potential values (RPM), and Supplementary Figs. 3c, 6b, e, j and 8c are provided as a Source Data file.

## Code availability
The MATLAB codes that were used to represent MEA raster and PSTH data are available from the corresponding author upon reasonable request.

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

## Acknowledgements

We thank Thierry Léveillard for providing the rd1 mice and Cheryl Craft for providing the hCAR antibody. We are thankful to Romain Caplette, Olivier Marre and Stéphane Deny for their help with the MEA recordings and analysis. We thank Abhishek Sengupta for the construction of the light/dark box. We are grateful to Mélissa Desrosiers and Camille Robert for AAV productions and the fundraising department of the Vision Institute for LCL fundraising. This study was supported by a ERC Starting Grants (OPTOGENRET, 309776/JD, REGNETHER 639888/DD), the Centre National de la Recherche Scientifique (CNRS), the Institut National de la Santé et de la Recherche Médicale (INSERM), Labex-Lifesenses (D.D., J.D.), Sorbonne Université, Marie Curie CIG (334130, Retinal Gene Therapy, D.D.), INSERM, ANR grant RHU Light4Deaf (D.D.,O.G.), LCL Foundation (D.D.), Deutsche Forschungsgemeinschaft (DFG) FZT 111, Center for Regenerative Therapies Dresden, FZT 111 Cluster of Excellence (M.A.), DFG Grant AD375/6-1 (M.A.), BMBF Research Grant 01EK1613A (M.A.).

## Author contributions

M.G. optimized differentiation and AAV. mediated transduction of hiPSC-derived retinal organoids, performed culture, histology, imaging, qPCR, designed experiments and wrote the manuscript. M.L. performed cell transplantation, imaging, histology, cell quantification, MEA recordings, behavioural experiments, designed experiments and wrote the manuscript. A.C. performed patch-clamp recordings and 2-photon imaging. L.G. generated hiPSC-derived retinal organoids, performed cell transplantation, imaging, histology and behavioural experiments. F.R. performed behavioural experiments. T.F. contributed to cell transplantation. S.G. and O.B. performed Y-chromosome FISH experiments and quantification, cell transplantation, histology and imaging. G.G. and S.R. helped to optimize hiPSC cultures and differentiation protocols. S.P. and J.A.S. provided scientific input, financial and administrative support. O.G. provided hiPSCs and gave feedback on the manuscript. M.A. provided Cpfl1/Rho$^{-/-}$ mice, contributed to cell transplantation and gave feedback on the manuscript. D.D. and J.D. designed experiments and wrote the manuscript.

## Additional information

**Competing interests:** Sorbonne Universite, Centre National De La Recherche Scientifique and INSERM have submitted a patent application (PCT/EP2017/074125) based on the work covered in this manuscript. J.D., M.A., A.C., M.G.-H., M.L., D.D., O.G. and J.-A.S are affiliated with these organisations and the patent. The remaining authors declare no competing interests.

