## [Peer Review File · Nature Communications]

Reviewers' Comments:

Reviewer #1:

Remarks to the Author:

This is a novel approach to the problem. The manuscript clearly presented. The study has a unique approach that combines optogenetics with retinal cell transplantation as a therapy for blindness. The authors should explain why implanting optogenetic photoreceptors is better than adding an optogenetic channel/GPCR/pump to any of the remaining retinal cells (like bipolar or ganglion). If the photoreceptor transplant isn't going to make Outer segments and the phototransduction cascade components, then why go through all the trouble of implanting them? Can one not add the same optogenetic effector to any surviving retinal cell. The approach as described seems more invasive than gene transfer of the optogenetic factor with no obvious advantage. They never address this issue or discuss the other groups strategies to add light sensitivity to surviving neurons at all. Further, in lines 212-213 they say, "we went beyond the current limitation of an optogenetic gene therapy approach, which can only rescue the function of remaining 'dormant' cones." That is just not true. I think they meant "current limitation of a photoreceptor directed optogenetic gene therapy approach."

The MEA data is a little weak. They showed that they can achieve up to 25 Hz firing rate when patched to these implanted photoreceptors, but they never give any similar MEA data. So it remains to be seen if the retina as a whole can rapidly respond to light.

The authors should clarify why they used eNpHR for the first half of the investigation and then used JAWS for the second half.

Reviewer #2:

Remarks to the Author:

In this study the authors describe the generation of optogenetically-transformed photoreceptors and their transplantation into mouse models of retina degeneration. First, the study describes the introduction of a hyperpolarizing microbial opsin (*Natronomonas pharaonis* halorhodopsin, NpHR) into photoreceptor precursors from new-born mice by AAV infection, leading to expression of NpHR and YFP under the control of the rhodopsin promoter; and transplantation of the optogenetically-transformed photoreceptor precursors into Cpf1/Rho-/- and C3H rd/rd (rd1) mice, followed by evaluation of light-response at the photoreceptor and ganglion cell level through patch-clamp recordings and MEA. Second, the study describes a similar AAV-based approach for the generation of optogenetically-transformed cone photoreceptors derived from human iPS cells, in this case expressing the chloride pump Jaws and GFP under the control of cone arrestin promoter; transplantation into Cpf1/Rho-/- and C3H rd/rd (rd1) mice, and evaluation of light response following the same approach as for the previous experiments.

The study concludes that the results from these experiments demonstrate that transplantation of optogenetically-transformed photoreceptor cells leads to vision restoration through structural and functional retinal repair. However, there are several important concerns that need to be addressed in order to fully support the main conclusions of this study.

Major concerns

One of the main concerns is that the study does not address several key issues for accurate evaluation and interpretation of outcomes after photoreceptor transplantation.

1. The study does not provide appropriate evaluation/evidence regarding the percentage and type of host photoreceptors remaining in the degenerating retina at the time of photoreceptor transplantation. The two models used in this study show progressive degeneration of the ONL over several weeks but with a percentage of photoreceptor cells remaining for a significant period of time afterwards (Santos-Ferreira et al, 2016; Carter-Dawson et al, 1978). The presence of remaining host photoreceptors at the time of transplantation is of critical importance in the context

of the already well documented mechanism of material transfer that occurs between transplanted and host photoreceptors. The current study does not properly address the issue of potential material transfer.

- The exact time of photoreceptor transplantation is not indicated. According to the information provided, in the case of Cpf1/Rho-/- mice they were transplanted some time between 5 to 18 weeks old, and in the case of C3H rd/rd (rd1) mice between 4 to 11 weeks. Of significant importance, the percentage of remaining host photoreceptors within this time frame is variable and in some cases still highly significant (for example Cpf1/Rho-/- mice 5-8 weeks old still have half-to a third of the ONL present, Santos-Ferreira et al, 2016).
- Figure 1 compares age-matched non-transplanted vs transplanted animals. This is useful as a way of visualizing the transplanted cells, but does not provide a comparison between the state of the ONL at the time of transplantation vs the time of analysis after transplantation. It would be important to document the composition of the ONL at the exact time of transplantation in order to better and more accurately evaluate the effects and outcomes of photoreceptor transplantation. This is of critical importance to either document or rule out a possible rescue effect of remaining host photoreceptors.
- The study should include specific, and properly designed assays addressing the possibility of material transfer or possible alternative mechanisms of rescue. Even at what is referred as end-stage degeneration in these models, there are remaining photoreceptor cells that although severely compromised and unable to elicit a measurable response, could be functionally rescued upon transplantation (Wang W et al, 2016).
- In the experiments involving transplantation of optogenetically-transformed photoreceptor precursors from new-born mice, the issue of material transfer has not been addressed in any way. This should be specifically addressed.
- In the experiments involving transplantation of optogenetically-transformed hiPSC-derived photoreceptors, the study includes immunohistochemical detection of transplanted cells from human origin with HNA and comparative measurements of nuclear size between human vs host cells. Even though these analyses could give initial support to the lack of material transfer they do not completely rule it out. At a minimum, the experiments should also include a thorough quantitative analysis of HNA+/GFP+ vs HNA-/GFP+ cells. As for the comparative measurements of nuclear size, averaging the measurements (as it has been done in this study) the majority of transplanted human photoreceptors would certainly mask any possible host photoreceptor expressing YFP due to material transfer.

2. The study does not include any analysis comparing the response from transplanted vs host photoreceptors. Within the context of the possibility of an underlying mechanisms of material transfer, this is of particular importance in view of the results reported by Busskamp et al, 2010. In the referenced study the authors demonstrated that expression of NpHR via AAV infection in surviving photoreceptors of late stage degeneration RD mice was enough to reactivate retinal ON and OFF pathways and enable RD mice to perform visually guided behaviors. This therefore opens the question of whether the light-responses recorded in the current study are indeed specifically driven by transplanted photoreceptors vs reactivated host photoreceptors.

3. The study does not provide appropriate evidence/evaluation of the percentage of surviving/integrated photoreceptors after transplantation. The number of YFP+ photoreceptors observed after transplantation in Figure 1 or GFP+ photoreceptors in Figure 4 seems rather low to support a significant increase in visual function as reported in this study.

4. The study does not provide conclusive evidence of structural and functional integration of transplanted photoreceptors. It would be important to provide conclusive demonstration of structural integration by a more robust documentation of the establishment of functional synaptic connections between transplanted photoreceptors and host bipolar cells.

5. There are additional important concerns regarding the electrophysiological recordings.

- The authors show (Fig 1H) that light evokes about 5 mV hyperpolarization in "NpHR

photoreceptors" as one would expect if a light-activated chloride pump works. However, the summary figure in Fig 1H bottom right shows instead 10-20 mV average responses.

- The ECl calculated in these experiments (based on the ionic composition of the pipette solution and the Ames used for bath) is about -57 mV, which would be consistent with the reported light-evoked 10-20 mV hyperpolarizations (which would be possible if the Vm in the dark is about -40 mV). Unfortunately, the study does not report the resting Vm of "NpHR photoreceptors" in the dark (at 0 current, as claimed on pg 18), for example for the trace shown in Fig 1H (bottom left traces) Fig 1I and in Fig 1J. Similarly, what was the resting Vm for the cell shown in Fig 3L, Fig 4F and Fig 4G?
- This is very critical, as light-evoked hyperpolarization of photoreceptors only conveys information to the rest of the circuit if it is associated with the reduction of glutamate release from the synaptic terminal of photoreceptors onto the dendrites of second-order retinal neurons, and most importantly, onto bipolar cells. As such, photoreceptors of any origin, must be depolarized to about -40 mV (or preferably even more depolarized) in the dark so that the (L-type) voltage-gated calcium channels at their axon terminals are open, and glutamate is being released. Rods and cones are depolarized by the cGMP-gated cation current in the dark, the channels sitting mostly in the outer segment. So, once again, was the Vm of "NpHR photoreceptors" or Jaw-expressing hiPSC-driven photoreceptors at or above -40 mV in dark? If that was the case, what channel (current) kept them depolarized?
- As mentioned before, the study does not report the exact age of the animals at the time of transplantation. The data presented in support of transplanted-photoreceptor-driven light response in Fig 2 and Fig 4, has been obtained from Cpf11/Rho-/- mice, which according to the manuscript were transplanted some time between 5 to 18 weeks old. As also mentioned above, the percentage of remaining host photoreceptors within this time frame is variable and, very importantly, still highly significant between week 5-10 (Santos-Ferreira et al, 2016). Still after 12 weeks, there are a significant number of cones remaining in the ONL. These issues raise important concerns regarding the light-responses recorded at the level of the ganglion cells, since it is likely these responses could be triggered by surviving original photoreceptors, as well as the light-driven behavior observed. Interestingly, it has been reported that ganglion cell responses can still be elicited from the non-transplanted late stage degenerating rd1 retina (Fujii et al, 2016).

We thank the reviewers for their comments that helped improve this manuscript. We have addressed all of the raised questions, as described below, by adding new experiments and new analysis and/or by discussing the issues that raised reviewers' concerns in the text. We have highlighted changes in the text in yellow and appended below the list of additional experiments and figures. Sylvia Gasparini and Oliver Borsch contributed critically to the newly added results and have therefore been added as authors.

The list of changes in figures and tables:

Figure 1 now adds data collected from the Y chromosome FISH probing, pointing to a low level of material transfer events between transplanted mouse NpHR-expressing photoreceptors and endogenous host photoreceptors (<10%; Fig. 1H, I).

In **Figure 3**, we removed all data obtained from mice transplanted with NpHR-expressing rods before the age of 9 weeks, for MEA data as well as the light/dark box test.

Figure S2 has been added, showing synaptic marker staining for *Cpfl1/Rho*^{-/-} mice transplanted with NpHR-expressing precursors.

Figure S3 shows data on YFP⁺ cell quantification 4 weeks after transplantation in *Cpfl1/Rho*^{-/-} mice.

Figure S8 adds quantification of HNA⁺/GFP⁺ versus HNA⁻/GFP⁺ photoreceptors.

We added **Table S4** that consists of a list all mice used to generate figures, showing their ages at the time of transplantation and at the time of experiment.

Reviewer #1 (Remarks to the Author):

This is a novel approach to the problem. The manuscript clearly presented. The study has a unique approach that combines optogenetics with retinal cell transplantation as a therapy for blindness.

The authors should explain why implanting optogenetic photoreceptors is better than adding an optogenetic channel/GPCR/pump to any of the remaining retinal cells (like bipolar or ganglion). If the photoreceptor transplant isn't going to make outer segments and the phototransduction cascade components, then why go through all the trouble of implanting them? Can one not add the same optogenetic effector to any surviving retinal cell. The approach as described seems more invasive than gene transfer of the optogenetic factor with no obvious advantage. They never address this issue or discuss the other group's strategies to add light sensitivity to surviving neurons at all. Further, in lines 212-213 they say, "we went beyond the current limitation of an optogenetic gene therapy approach, which can only rescue the function of remaining 'dormant' cones." That is just not true. I think they meant "current limitation of a photoreceptor directed optogenetic gene therapy approach."

We thank the reviewer for these positive comments. Indeed, optogenetics to restore light sensitivity in blind retinas has been attempted multiple times and is considered a promising gene therapy strategy to restore vision with two ongoing clinical trials targeting microbial opsins to RGCs. However, our approach is preferable for several reasons.

Conferring light sensitivity to cells lying downstream from photoreceptors¹⁻⁸ bypasses the information processing normally conducted by the retinal circuitry. For example, studies in which light sensitive proteins were expressed in RGCs lost both the processing occurring in the OPL as well as the IPL, resulting in recovery of only ON RGC responses. In this approach, the retina might be missing the type of image pre-processing needed to achieve optimal vision. A further difficulty with this approach is that only a ring of RGCs around the fovea can be targeted with AAV vectors in primate species. Since microbial opsins require high light intensities and require stimulation goggles, this also requires the use of a complex light stimulation device directly stimulating the transfected ring-shaped RGC area.

Targeting bipolar cells at the least allows keeping the IPL processing, and both ON and OFF responding RGCs were documented in some of these studies ^{5,7,9}. However, complex visual functions such as lateral inhibition and directional selectivity were only recovered when the response stemmed from the photoreceptor layer ¹⁰. Our approach potentially allows recovery of sophisticated visual functions that cannot be recovered when conferring light sensitivity to bipolar cells or RGCs. Moreover, in the case of the bipolar cells, their number and density is fairly small and targeting them with AAV vectors has only been achieved in the marmoset retina. Thus the very limited number of bipolar cells that can be targeted will further limit the resolution of vision in contrast to a much larger area that can be covered with transplantation of a high number of photoreceptors.

Lastly, optogenetics has also been implemented in the so-called 'dormant cones' ^{10,11}. Indeed, this was the only time where resensitized photoreceptors activated all retinal cone pathways; drove sophisticated retinal circuit functions (including directional selectivity), activated cortical circuits, and mediated visually guided behaviors all at once. Indeed it was demonstrated that persisting cone cell bodies (~25%) were enough to induce ganglion cell activity, even during later stages of degeneration in mouse models but it was uncertain at that point what percentage of retinitis pigmentosa patients still maintained dormant cones. Later studies conducted at our clinical investigation center showed this to be around 17% in a population of patients with average age of 56 ¹². For patients who have already gone beyond the loss of photoreceptors, transplantation of functional cones therefore offers all of the advantages of photoreceptor targeted optogenetics, particularly in regard to full retinal processing after light stimulation.

This point is now added and discussed in the revised manuscript.

The MEA data is a little weak. They showed that they can achieve up to 25 Hz firing rate when patched to these implanted photoreceptors, but they never give any similar MEA data. So it remains to be seen if the retina as a whole can rapidly respond to light.

We agree with the reviewer that it is not because photoreceptors can follow up to 25 Hz stimulation that downstream ganglion cells will also be able to do so. In fact,

previous literature has shown that high frequencies (above 10Hz) are filtered out already at the stage of bipolar cells¹³. Our MEA data thus do not imply that ganglion cells will be able to follow a 25-Hz stimulation. We acknowledged and clarified this point in the revised text.

The authors should clarify why they used eNpHR for the first half of the investigation and then used JAWS for the second half.

We used two different proteins because our work on donor-derived photoreceptor precursors started before Jaws was described¹¹ and NpHR performed quite well in mouse photoreceptors. However, NpHR is difficult to express at the cell membrane in human cells¹⁴, likely because the trafficking signals used to engineer these bacterial proteins originated from rodent sequences¹⁴⁻¹⁶. Moreover, Jaws shows better response amplitudes compared to NpHR¹¹. For all of these reasons, we decided to transition to Jaws for the second part of the study using cells of human origin. We have now made this clear in the manuscript text.

Reviewer #2 (Remarks to the Author):

In this study the authors describe the generation of optogenetically-transformed photoreceptors and their transplantation into mouse models of retina degeneration. First, the study describes the introduction of a hyperpolarizing microbial opsin (Natronomonas pharaonis halorhodopsin, NpHR) into photoreceptor precursors from new-born mice by AAV infection, leading to expression of NpHR and YFP under the control of the rhodopsin promoter; and transplantation of the optogenetically-transformed photoreceptor precursors into Cpf1/Rho-/- and C3H rd/rd (rd1) mice, followed by evaluation of light-response at the photoreceptor and ganglion cell level through patch-clamp recordings and MEA. Second, the study describes a similar AAV-based approach for the generation of optogenetically-transformed cone photoreceptors derived from human iPS cells, in this case expressing the chloride pump Jaws and GFP under the control of cone arrestin promoter; transplantation into Cpf1/Rho-/- and C3H rd/rd (rd1) mice, and evaluation of light response following the same approach as for the previous experiments.

The study concludes that the results from these experiments demonstrate that transplantation of optogenetically-transformed photoreceptor cells leads to vision restoration through structural and functional retinal repair. However, there are several important concerns that need to be addressed in order to fully support the main conclusions of this study.

Major concerns

One of the main concerns is that the study does not address several key issues for accurate evaluation and interpretation of outcomes after photoreceptor transplantation.

1. The study does not provide appropriate evaluation/evidence regarding the percentage and type of host photoreceptors remaining in the degenerating retina at the time of photoreceptor transplantation. The two models used in this study show

progressive degeneration of the ONL over several weeks but with a percentage of photoreceptor cells remaining for a significant period of time afterwards (Santos-Ferreira et al, 2016; Carter-Dawson et al, 1978). The presence of remaining host photoreceptors at the time of transplantation is of critical importance in the context of the already well documented mechanism of material transfer that occurs between transplanted and host photoreceptors. The current study does not properly address the issue of potential material transfer.

We thank the reviewer for this pertinent comment. However, quantification of remaining photoreceptors for the *Cpfl1/Rho*^{-/-} model has been conducted in the study mentioned by the reviewer ¹⁷. It shows about 2-3 rows of remaining photoreceptors at 9 weeks, the earliest time point for our transplantations.

Photoreceptor degeneration in the rd1 mouse starts at around P8, and by 3 weeks of age, there are no outer segments (OS) and only a single row of cell bodies remaining in the ONL, consisting of predominantly cone photoreceptors ¹⁸⁻²¹. These are subsequently lost through a secondary mechanism. However, non-functional, dormant cones have been detected in rd1 mice for prolonged periods of time – even after more than 250 days ¹⁰.

We agree with the reviewer that the existence of remaining photoreceptors raises the possibility of material transfer. We therefore included new experiments in which NpHR/YFP-photoreceptor precursors from male donors were transplanted into female hosts with residual ONL (*Cpfl1/Rho*^{-/-} at 9 weeks of age) and used Y chromosome FISH probing to distinguish cytoplasmic and nuclear labeling and thus assessed cytoplasmic exchange ²². Our data shows that over 90% of photoreceptors that stain positive for NpHR/YFP also stain positive for Y chromosome, confirming that the vast majority of NpHR/YFP⁺ cells are of donor origin. We added these data to the revised Figure 1 and discussed the potential involvement of material transfer to functional improvements in further detail in the discussion. We agree that our recent FISH results do not fully rule out that material transfer contributes to the improved functional responses, but we suggest that it is minor, if present at all, based on the following observations:

1. The level of functional improvement is independent of the host age at time of injection, i.e. no differences were observed between 9 and 18 weeks despite significant differences in the number of remaining endogenous photoreceptors.

2. Material transfer is rare between human donor and rodent host photoreceptors^{23,24} (confirmed by our own results shown in Figure 5 and S8), arguing against a significant contribution of material transfer to the observed functional improvements.

• *The exact time of photoreceptor transplantation is not indicated. According to the information provided, in the case of Cplf1/Rho^{-/-} mice they were transplanted sometime between 5 to 18 weeks old, and in the case of C3H rd/rd (rd1) mice between 4 to 11 weeks. Of significant importance, the percentage of remaining host photoreceptors within this time frame is variable and in some cases still highly significant (for example Cplf1/Rho^{-/-} mice 5-8 weeks old still have half-to a third of the ONL present, Santos-Ferreira et al, 2016).*

We understand the concerns raised by the reviewer about mice transplanted at a younger age, and therefore decided to exclude all data for Cplf1/Rho^{-/-} mice transplanted prior to 9 weeks of age. Moreover, we added a supplementary table with exact ages of all mice included in our figures to the revised manuscript (Table S4).

• *Figure 1 compares age-matched non-transplanted vs transplanted animals. This is useful as a way of visualizing the transplanted cells, but does not provide a comparison between the state of the ONL at the time of transplantation vs the time of analysis after transplantation. It would be important to document the composition of the ONL at the exact time of transplantation in order to better and more accurately evaluate the effects and outcomes of photoreceptor transplantation. This is of critical importance to either document or rule out a possible rescue effect of remaining host photoreceptors.*

The composition of the ONL of Cplf1/Rho^{-/-} mice and rd1 mice at different time points has been documented in previous publications^{10,17,18}. However, we can rule out any significant rescue effect mediated by remaining host photoreceptors as our control groups transplanted at the same ages with wild type donor-derived photoreceptor

precursors or hiPSC-derived photoreceptors expressing GFP only did not show any functional response (Figure 2- 5, Figure S5 and S10). This confirms that any possible rescue effect on remaining host photoreceptors cannot be a result of the transplantation itself but the rescue effect is directly correlated to the presence of an optogenetic protein expressed in the transplanted photoreceptors.

- *The study should include specific, and properly designed assays addressing the possibility of material transfer or possible alternative mechanisms of rescue. Even at what is referred as end-stage degeneration in these models, there are remaining photoreceptor cells that although severely compromised and unable to elicit a measurable response, could be functionally rescued upon transplantation (Wang W et al, 2016).*

We agree with the reviewer on the importance of potential rescue via material transfer and have conducted Y chromosome FISH experiments for mouse to mouse transplantation alongside quantification of cells positive for Jaws and displaying the HNA human marker in relation to host photoreceptors as detailed above and below. As for functional rescue of remaining photoreceptors by other mechanisms, such as those described in Wang et al., 2016²⁵; we cannot rule them out completely however we are confident that if there were improved functional responses in these remaining photoreceptors it would be minor as the level of functional improvement is independent of the host age at time of injection, i.e. no differences between 9 and 18 weeks despite significant differences in the number of remaining endogenous photoreceptors. Again, there are no such effects observed when GFP-expressing cells are transplanted, further supporting that such rescue effects – if present- are minor or undetectable. In addition, the functional tests were performed in conditions which specifically activate the optogenetic protein, thus it is unlikely that responses are due to neuroprotective effects on endogenous photoreceptors.

- *In the experiments involving transplantation of optogenetically-transformed photoreceptor precursors from new-born mice, the issue of material transfer has not been addressed in any way. This should be specifically addressed.*

The question of material transfer is addressed in the revised version of the manuscript by the Y chromosome FISH method. NpHR/YFP-expressing rod

precursors derived from male P4 mice were injected into female *Cpfl1/Rho*^{-/-} mice at 9 weeks of age, and imaged after 4 weeks using structured illumination microscopy. Y chromosome⁺/YFP⁺ cells (transplanted donor cells) and Y chromosome⁻/YFP⁺ cells (endogenous photoreceptor that underwent material transfer) were quantified. More than 90% of YFP⁺ cells co-stained with the Y chromosome probe, leaving only very few cells exclusively YFP⁺. This could either be due to an artefact or very rare events of cytoplasmic exchange among donor and host photoreceptors. The results are presented in the revised Figure 1 and further discussed above and in the revised text.

• In the experiments involving transplantation of optogenetically-transformed hiPSC-derived photoreceptors, the study includes immunohistochemical detection of transplanted cells from human origin with HNA and comparative measurements of nuclear size between human vs host cells. Even though these analyses could give initial support to the lack of material transfer they do not completely rule it out. At a minimum, the experiments should also include a thorough quantitative analysis of HNA⁺/GFP⁺ vs HNA⁻/GFP⁺ cells. As for the comparative measurements of nuclear size, averaging the measurements (as it has been done in this study) the majority of transplanted human photoreceptors would certainly mask any possible host photoreceptor expressing YFP due to material transfer.

Following the reviewer's suggestion, we quantified the number of HNA⁺/GFP⁺ and HNA⁻/GFP⁺ cells in retinas transplanted with hiPSC-derived cones (N=3). The results are presented in Figure S8. Only very few HNA⁻/GFP⁺ cells were detected (5%), suggesting that the occurrence of material transfer was extremely limited. This is in line with previous observations in rodent models of outer retinal degeneration^{23,24}. We believe that such limited instances of material transfer are highly unlikely to contribute to the functional improvements as supported by our results in young versus old hosts and the lack of rescue with GFP-expressing cells.

*2. The study does not include any analysis **comparing the response from transplanted vs host photoreceptors**. Within the context of the possibility of an underlying mechanisms of material transfer, this is of particular importance in view of the results reported by Busskamp et al, 2010. In the referenced study the authors*

demonstrated that expression of NpHR via AAV infection in surviving photoreceptors of late stage degeneration RD mice was enough to reactivate retinal ON and OFF pathways and enable RD mice to perform visually guided behaviors. This therefore opens the question of whether the light-responses recorded in the current study are indeed specifically driven by transplanted photoreceptors vs reactivated host photoreceptors.

As described in a response to one of the previous comments, Y chromosome FISH experiment has been added to the revised version of the manuscript, showing less than 10% of labelled cells seen in the subretinal space following transplantation of mouse rod precursors were potentially host cells that underwent material transfer (Figure 1). In addition, HNA⁺/GFP⁺ and HNA⁻/GFP⁺ have been quantified 4 weeks after transplantation of hiPSC-derived cones. The results indicated that only 5% of fluorescent cells were possibly endogenous mouse cells (Figure S8). This is now stated in the new version of the manuscript.

3. The study does not provide appropriate evidence/evaluation of the percentage of surviving/integrated photoreceptors after transplantation. The number of YFP⁺ photoreceptors observed after transplantation in Figure 1 or GFP⁺ photoreceptors in Figure 4 seems rather low to support a significant increase in visual function as reported in this study.

We agree with the reviewers' remark. Indeed, more cells than can be visualized in the initial images usually survive in the subretinal space. We thus performed cell counts of surviving YFP⁺ cells from 6 *Cpfl1/Rho*^{-/-} retinas transplanted with donor-derived NpHR-expressing rod precursors and included this data set as a new supplementary figure (Figure S3).

*4. The study does not provide conclusive evidence of structural and functional integration of transplanted photoreceptors. It would be important to provide conclusive demonstration of structural integration by a more robust documentation of the establishment of functional synaptic **connections** between transplanted photoreceptors and host bipolar cells.*

In agreement with the reviewer's comment we have added this missing data to strengthen our study. We performed additional immunohistochemical staining for the synaptic marker synaptophysin in NpHR-rod precursor-transplanted *Cpfl1/Rho*^{-/-} mice. These are shown in Figure S2.

5. There are additional important concerns regarding the electrophysiological recordings.

• The authors show (Fig 1H) that light evokes about 5 mV hyperpolarization in "NpHR photoreceptors" as one would expect if a light-activated chloride pump works. However, the summary figure in Fig 1H bottom right shows instead 10-20 mV average responses.

Light evokes a ~5 mV response on fig1H example trace, and the summary does show a mean value of 10.4 ± 2.9 (mean \pm SEM, n=6). Indeed cells' response amplitude varied considerably (between 2 and 18.5 mV) for the different NpHR-photoreceptor-transplanted to *Cpfl1/Rho*^{-/-} mice that we recorded, and the 5 mV example we chose to present was displaying a good signal to noise ratio.

• The ECl calculated in these experiments (based on the ionic composition of the pipette solution and the Ames used for bath) is about -57 mV, which would be consistent with the reported light-evoked 10-20 mV hyperpolarizations (which would be possible if the Vm in the dark is about -40 mV). Unfortunately, the study does not report the resting Vm of "NpHR photoreceptors" in the dark (at 0 current, as claimed on pg 18), for example for the trace shown in Fig 1H (bottom left traces) Fig 1I and in Fig 1J. Similarly, what was the resting Vm for the cell shown in Fig 3L, Fig 4F and Fig 4G?

We calculated the mean resting Vm of both NpHR-photoreceptors and Jaws-expressing hiPSC-driven photoreceptors (at 0 current, dark adapted). The mean value for transplanted NpHR photoreceptors (Figure 2) RMP was -36.2 ± 1.5 mV (mean \pm SEM, n=4) and for transplanted Jaw-expressing hiPSC-driven photoreceptors (Figure 5) RMP was -40.8 ± 5.2 mV (mean \pm SEM, n=5). For Jaws-

expressing cells within organoids (Figure 4) RMP was -41.7 ± 3.9 mV (mean \pm SEM, n=4).

- *This is very critical, as light-evoked hyperpolarization of photoreceptors only conveys information to the rest of the circuit if it is associated with the reduction of glutamate release from the synaptic terminal of photoreceptors onto the dendrites of second-order retinal neurons, and most importantly, onto bipolar cells. As such, photoreceptors of any origin, must be depolarized to about -40 mV (or preferably even more depolarized) in the dark so that the (L-type) voltage-gated calcium channels at their axon terminals are open, and glutamate is being released. Rods and cones are depolarized by the cGMP-gated cation current in the dark, the channels sitting mostly in the outer segment. So, once again, was the V_m of “NpHR photoreceptors” or Jaw-expressing hiPSC-driven photoreceptors at or above -40 mV in dark? If that was the case, what channel (current) kept them depolarized?*

As for the concern about the physiological state of the transplanted photoreceptors, we calculated the mean resting V_m of both NpHR-expressing mouse photoreceptors and Jaws-expressing hiPSC-derived photoreceptors in dark adapted conditions at 0 current state, and found them to be depolarized between -35 to -40 mV eliminating any concern about their potential ability to release glutamate. We find that this is coherent with the results reported by Buskamp et al., 2010¹⁰, where degenerated cone cell bodies completely lacking outer segments were found to be surprisingly depolarized at -26 mV. However, we believe finding the exact mechanistic explanation for these depolarized values is beyond the scope of our study.

- *As mentioned before, the study does not report the exact age of the animals at the time of transplantation. The data presented in support of transplanted-photoreceptor-driven light response in Fig 2 and Fig 4, has been obtained from *Cpfl1/Rho-/-* mice, which according to the manuscript were transplanted sometime between 5 to 18 weeks old. As also mentioned above, the percentage of remaining host photoreceptors within this time frame is variable and, very importantly, still highly significant between week 5-10 (Santos-Ferreira et al, 2016). Still after 12 weeks, there are a significant number of cones remaining in the ONL. **These issues raise important concerns regarding the light-responses recorded at the level of the***

ganglion cells, since it is likely these responses could be triggered by surviving original photoreceptors, as well as the light-driven behavior observed. Interestingly, it has been reported that ganglion cell responses can still be elicited from the non-transplanted late stage degenerating rd1 retina (Fuji et al, 2016).

We excluded all data obtained from treated *Cpfl1/Rho^{-/-}* mice that were injected younger than 9 weeks old. We also added a Supplementary table showing the ages of all mice included in figures (Table S4). We are acknowledging the reviewer's concern about possible RGC responses in late stage degeneration. However, we have not detected any light responses from RGCs in the recorded controls, i.e. in retinas transplanted with GFP only expressing precursors, retinas transplanted with GFP only hiPSC-derived cones or non-transplanted retinas.

List of references

- 1 Bi, A. *et al.* Ectopic expression of a microbial-type rhodopsin restores visual responses in mice with photoreceptor degeneration. *Neuron* **50**, 23-33, doi:10.1016/j.neuron.2006.02.026 (2006).
- 2 Sengupta, A. *et al.* Red-shifted channelrhodopsin stimulation restores light responses in blind mice, macaque retina, and human retina. *EMBO molecular medicine* **8**, 1248-1264, doi:10.15252/emmm.201505699 (2016).
- 3 Berry, M. H. *et al.* Restoration of patterned vision with an engineered photoactivatable G protein-coupled receptor. *Nature communications* **8**, 1862, doi:10.1038/s41467-017-01990-7 (2017).
- 4 Zhang, Y., Ivanova, E., Bi, A. & Pan, Z. H. Ectopic expression of multiple microbial rhodopsins restores ON and OFF light responses in retinas with photoreceptor degeneration. *The Journal of neuroscience : the official journal of the Society for Neuroscience* **29**, 9186-9196, doi:10.1523/JNEUROSCI.0184-09.2009 (2009).
- 5 Cronin, T. *et al.* Efficient transduction and optogenetic stimulation of retinal bipolar cells by a synthetic adeno-associated virus capsid and promoter. *EMBO molecular medicine* **6**, 1175-1190, doi:10.15252/emmm.201404077 (2014).
- 6 Lagali, P. S. *et al.* Light-activated channels targeted to ON bipolar cells restore visual function in retinal degeneration. *Nature neuroscience* **11**, 667-675, doi:10.1038/nn.2117 (2008).
- 7 Mace, E. *et al.* Targeting channelrhodopsin-2 to ON-bipolar cells with vitreally administered AAV Restores ON and OFF visual responses in blind mice. *Molecular therapy : the journal of the American Society of Gene Therapy* **23**, 7-16, doi:10.1038/mt.2014.154 (2015).
- 8 Chaffiol, A. *et al.* A New Promoter Allows Optogenetic Vision Restoration with Enhanced Sensitivity in Macaque Retina. *Molecular therapy : the journal of the American Society of Gene Therapy* **25**, 2546-2560, doi:10.1016/j.ymthe.2017.07.011 (2017).

- 9 van Wyk, M., Pielecka-Fortuna, J., Lowel, S. & Kleinlogel, S. Restoring the ON Switch in Blind
Retinas: Opto-mGluR6, a Next-Generation, Cell-Tailored Optogenetic Tool. *PLoS biology* **13**,
e1002143, doi:10.1371/journal.pbio.1002143 (2015).
- 10 Busskamp, V. *et al.* Genetic reactivation of cone photoreceptors restores visual responses in
retinitis pigmentosa. *Science* **329**, 413-417, doi:10.1126/science.1190897 (2010).
- 11 Chuong, A. S. *et al.* Noninvasive optical inhibition with a red-shifted microbial rhodopsin.
Nature neuroscience **17**, 1123-1129, doi:10.1038/nn.3752 (2014).
- 12 Azoulay-Sebban, L. *Etude des corrélations anatomiques et fonctionnelles au cours de la*
rétinopathie pigmentaire : identification et validation de nouveaux marqueurs prédictifs
Sciences Recherche Clinique thesis, Paris 6, (2015).
- 13 Crevier, D. W. & Meister, M. Synchronous period-doubling in flicker vision of salamander
and man. *Journal of neurophysiology* **79**, 1869-1878, doi:10.1152/jn.1998.79.4.1869 (1998).
- 14 Garita-Hernandez, M. *et al.* Optogenetic Light Sensors in Human Retinal Organoids. *Front*
Neurosci **12**, 789, doi:10.3389/fnins.2018.00789 (2018).
- 15 Gradinaru, V., Thompson, K. R. & Deisseroth, K. eNpHR: a Natronomonas halorhodopsin
enhanced for optogenetic applications. *Brain cell biology* **36**, 129-139, doi:10.1007/s11068-
008-9027-6 (2008).
- 16 Gradinaru, V. *et al.* Molecular and cellular approaches for diversifying and extending
optogenetics. *Cell* **141**, 154-165, doi:10.1016/j.cell.2010.02.037 (2010).
- 17 Santos-Ferreira, T. *et al.* Stem Cell-Derived Photoreceptor Transplants Differentially
Integrate Into Mouse Models of Cone-Rod Dystrophy. *Investigative ophthalmology & visual*
science **57**, 3509-3520, doi:10.1167/iovs.16-19087 (2016).
- 18 Carter-Dawson, L. D., LaVail, M. M. & Sidman, R. L. Differential effect of the rd mutation on
rods and cones in the mouse retina. *Investigative ophthalmology & visual science* **17**, 489-
498 (1978).
- 19 Drager, U. C. & Hubel, D. H. Studies of visual function and its decay in mice with hereditary
retinal degeneration. *The Journal of comparative neurology* **180**, 85-114,
doi:10.1002/cne.901800107 (1978).
- 20 Farber, D. B., Flannery, J. G. & Bowesrickman, C. The Rd Mouse Story - 70 Years of Research
on an Animal-Model of Inherited Retinal Degeneration. *Prog Retin Eye Res* **13**, 31-64, doi:Doi
10.1016/1350-9462(94)90004-3 (1994).
- 21 LaVail, M. M. & Sidman, R. L. C57BL-6J mice with inherited retinal degeneration. *Archives of*
ophthalmology **91**, 394-400 (1974).
- 22 Santos-Ferreira, T. *et al.* Retinal transplantation of photoreceptors results in donor-host
cytoplasmic exchange. *Nature communications* **7**, 13028, doi:10.1038/ncomms13028 (2016).
- 23 Gonzalez-Cordero, A. *et al.* Recapitulation of Human Retinal Development from Human
Pluripotent Stem Cells Generates Transplantable Populations of Cone Photoreceptors. *Stem*
cell reports **9**, 820-837, doi:10.1016/j.stemcr.2017.07.022 (2017).
- 24 Gagliardi, G. *et al.* Characterization and Transplantation of CD73-Positive Photoreceptors
Isolated from Human iPSC-Derived Retinal Organoids. *Stem cell reports* **11**, 665-680,
doi:10.1016/j.stemcr.2018.07.005 (2018).
- 25 Wang, W. *et al.* Two-Step Reactivation of Dormant Cones in Retinitis Pigmentosa. *Cell*
reports **15**, 372-385, doi:10.1016/j.celrep.2016.03.022 (2016).

Reviewers' Comments:

Reviewer #1:

None

Reviewer #2:

Remarks to the Author:

The authors have appropriately addressed all concerns regarding the original version of the manuscript. There are only minor editorial issues remaining that would be important to address as well.

Addition of Table S4 is very useful and provides important information in a very straight forward manner. Please double check the information in the table to make sure it is accurate; for example, the 3rd row indicates that Fig 1 E & F correspond to Cpf1Rho animals but the corresponding figure and figure legend indicate these correspond to rd1 mouse. Additionally, it is not clear the meaning of the column labeled as # of mice of same ages; most rows don't have any data on it and the ones that indicate the # of animals, the actual number doesn't seem to correlated with the numbers on the text and figure legends. animals o =f same age.

Please check Fig 1 I: y axis is labeled as % of GFP+ cells but the data corresponds to YFP cells - shmpiuld this be % of YFP+ cells instead?

In the methods section is indicated that sections for FISH and YFP+ quantification have been labeled with eGFP antibody instead of a YFP antibody. If this is just a type it should be corrected, but if eGFP Ab was actually used, there doesn't seem to be any explanation on what is that anywhere in the manuscript, so a clarification should be included.

Please include information on the calculated resting Vm (RMP) of photoreceptors in the dark (at 0 current) for the recordings presented in the figures. The calculations have been included in the rebuttal letter but I could not find the information not the manuscript, and it would be important to add this information to the final manuscript.

REVIEWERS' COMMENTS:

Reviewer #2 (Remarks to the Author):

The authors have appropriately addressed all concerns regarding the original version of the manuscript. There are only minor editorial issues remaining that would be important to address as well.

Addition of Table S4 is very useful and provides important information in a very straight forward manner. Please double check the information in the table to make sure it is accurate; for example, the 3rd row indicates that Fig 1 E & F correspond to Cpf1Rho animals but the corresponding figure and figure legend indicate these correspond to rd1 mouse. Additionally, it is not clear the meaning of the column labelled as # of mice of same ages; most rows don't have any data on it and the ones that indicate the # of animals, the actual number doesn't seem correlated with the numbers on the text and figure legends.

We apologize for the mistake in the table and we thank the reviewer for pointing it out. We have again verified the information in the table (now Supplementary Table 1) making sure it is accurate for all the figures' panels. The title of the last column has been changed to "Number of mice (N)" and N=1 has been added to rows that remained empty in the previous version of the table. The reason we added this row was to avoid the table becoming excessively long. For example, for Figure 1i, N=3, meaning that the data retrieved from 3 mice transplanted at the same age (9 weeks in this case) and used for experiments at the same age (13 weeks in this case) were included in the representation in Figure 1i. Please let us know if you still find this part of the table unclear, so we try to arrange it differently.

Please check Fig 1 I: y axis is labelled as % of GFP+ cells but the data corresponds to YFP cells - should this be % of YFP+ cells instead?

In the methods section is indicated that sections for FISH and YFP+ quantification have been labelled with eGFP antibody instead of a YFP antibody. If this is just a type it should be corrected, but if eGFP Ab was actually used, there doesn't seem to be any explanation on what is that anywhere in the manuscript, so a clarification should be included.

GFP antibody has indeed been used to amplify the YFP signal already present in transduced transplanted mouse-derived photoreceptors. In Figure 1i, and other figures where GFP antibody has been used to label NpHR-YFP⁺ cells, YFP has been changed to GFP. The fact that we used GFP antibody when performing immunohistochemical staining on NpHR-YFP⁺ transplanted animals is now clearly mentioned in Figure Legends and Supplementary Figure Legends where applicable.

Please include information on the calculated resting V_m (RMP) of photoreceptors in the dark (at 0 current) for the recordings presented in the figures. The calculations have been included in the rebuttal letter but I could not find the information not the manuscript, and it would be important to add this information to the final manuscript.

The information on the resting membrane potential (RMP) is now included in appropriate Figure Legends (for Figures 2, 4 and 5). The measurements leading to these values are also available in the Source data file.